

# Field theory of interacting boundary gravitons

Stephen Ebert[1], Eliot Hijano[2], Per Kraus[1], Ruben Monten[1] and Richard M. Myers[1]

**1** Mani L. Bhaumik Institute for Theoretical Physics
Department of Physics & Astronomy, University of California, Los Angeles, CA 90095, USA
**2** Department of Physics, Princeton University, Princeton, NJ 08544, USA

## Abstract

Pure three-dimensional gravity is a renormalizable theory with two free parameters labelled by $G$ and $\Lambda$. As a consequence, correlation functions of the boundary stress tensor in AdS$_3$ are uniquely fixed in terms of one dimensionless parameter, which is the central charge of the Virasoro algebra. The same argument implies that AdS$_3$ gravity at a finite radial cutoff is a renormalizable theory, but now with one additional parameter corresponding to the cutoff location. This theory is conjecturally dual to a $T\overline{T}$-deformed CFT, assuming that such theories actually exist. To elucidate this, we study the quantum theory of boundary gravitons living on a cutoff planar boundary and the associated correlation functions of the boundary stress tensor. We compute stress tensor correlation functions to two-loop order ($G$ being the loop counting parameter), extending existing tree level results. This is made feasible by the fact that the boundary graviton action simplifies greatly upon making a judicious field redefinition, turning into the Nambu-Goto action. After imposing Lorentz invariance, the correlators at this order are found to be unambiguous up to a single undetermined renormalization parameter.



# 1 Introduction

Three-dimensional gravity in the presence of a negative cosmological constant, as described by the Euclidean Einstein-Hilbert action supplemented with boundary terms

$$I = -\frac{1}{16\pi G} \int_M d^3x \sqrt{g}\big(R - 2\Lambda\big) + S_{\text{bndy}} \,, \tag{1}$$

is a perturbatively renormalizable theory since all candidate counterterms can be removed by field redefinitions [1]. The perturbative expansion around an AdS$_3$ background is well understood: one obtains a theory of boundary gravitons governed by Virasoro symmetry [2,3]. The quantum theory of these boundary gravitons is perfectly sensible and self-contained, with a well defined Hilbert space and spectrum of local operators. Indeed it is an extremely simple theory, as the action and stress tensor are rendered quadratic in appropriate field variables [4–6]. In CFT parlance, this theory describes the Virasoro vacuum block of some putative CFT with some spectrum of primary operators. A much studied problem is how to reconcile the desired modular invariance of such a spectrum with a sum over geometries interpretation in gravity, e.g. [1,3,7].

Besides introducing new states, another route to enriching and extending the theory of boundary gravitons is to radially move the AdS$_3$ boundary inwards, and there are several motivations for doing so. One is as a way to gain access to observables that are "more local" than the usual asymptotically defined quantities, namely the S-matrix in Minkowski space and boundary correlators in AdS. The need to develop such observables has long been appreciated, particularly in a cosmological context where there may not exist any "far away spatial region" that an observer at fixed time can appeal to. In general dimensions, the complications of defining quantum gravity in a finite spatial region is hard to disentangle from the usual UV problems,[1] but the situation is better in AdS$_3$ since the renormalizability argument applied to (1) applies also to the case of a finite boundary.[2] The problem is also interesting due to its proposed description [9] as a $T\overline{T}$-deformed CFT [10,11].[3] These are theories described in the IR as CFTs perturbed by irrelevant operators; their UV description is not well understood, but they conceivably represent a new type of quantum theory in which locality breaks down in a controlled manner. We take the perspective that these two descriptions — cutoff AdS$_3$ and $T\overline{T}$-deformed CFT — are mutually illuminating.

In this work, we develop the quantum theory of boundary gravitons on a cutoff planar surface, focusing on obtaining the optimal form of the action and using it to compute correlation functions of boundary operators. We now briefly summarize our findings. We work in the framework of the covariant phase space formalism [15,16], and in both the metric and Chern-Simons formulation [17,18] of 3D gravity, since they offer useful complementary perspectives. Our phase space is constructed by starting from an AdS$_3$ background and performing all coordinate/gauge transformations that preserve a Dirichlet boundary condition. Coordinates on this phase space can be taken to consist of two functions defined at some initial time on the boundary; these can be thought of as the coordinate transformations $(x,t) \to \big(x + A(x,t), t + B(x,t)\big)$ evaluated at $t = 0$. To construct the canonical formulation, we need a symplectic form and a Hamiltonian on this phase space, and we develop efficient methods for computing these. In the asymptotically AdS$_3$ case, this procedure is simple to carry out exactly, and we readily arrive at the Alekseev-Shatashvili action [4,5], as was obtained via the Chern-Simons formulation in [6]. At finite cutoff, life is more complicated; we work order-by-order in the $(A,B)$

---

[1]See [8] for a review of the boundary value problem in $D > 3$ Euclidean gravity.

[2]At least if the boundary is flat, as will be the main case of interest in this work. More generally, we might need boundary counterterms involving boundary curvature.

[3]At the classical level, this relation was substantiated in [12] using the perspective of mixed boundary conditions at infinity. See [13,14] for reviews of the $T\overline{T}$ deformation, its applications, and its relation to holography.

variables, but the resulting expressions quickly become complicated, in particular because the phase space action contains an ever growing number of higher derivatives acting on these fields.

A pleasant surprise (at least to us) is that a field redefinition $(A,B) \to (f, \overline{f})$ can be used to remove all higher derivatives from the action, at least to the order we have checked (eighth order in the fields). The resulting (imaginary time) action is none other than the Nambu-Goto action written in Hamiltonian form,[4]

$$I = \frac{1}{32\pi G} \int d^2 x \left[ i f' \dot{f} - i \overline{f}' \dot{\overline{f}} - 4 \frac{\sqrt{1 - \frac{1}{2} r_c (f'^2 + \overline{f}'^2) + \frac{1}{16} r_c^2 (f'^2 - \overline{f}'^2)^2} - 1}{r_c} \right], \quad (2)$$

where $r_c$ labels the radial location of the boundary such that $r_c \to 0$ is the asymptotic boundary. We obtain further evidence for this action by deriving it to all orders in the special case of linearly varying $(f, \overline{f})$. However, the stress tensor is not the canonical stress tensor of the Nambu-Goto theory due to the non-linear action of the Poincaré group on the fields. Rather, it includes a series of higher derivative correction terms reflecting the nonlocal nature of the theory, e.g.[5]

$$4G T_{zz} = \frac{1}{2} f'' - \frac{1}{4} f'^2 + \frac{1}{4} r_c f''' \overline{f}' - \frac{1}{8} r_c \left( f'^2 - 2 f' \overline{f}' \right)' \overline{f}' + \frac{1}{16} r_c^2 \left( f'''' \overline{f}'^2 + (f'^2)'' \overline{f}'' \right) + \dots. \quad (3)$$

Having obtained expressions for the action and stress tensor, we seek to quantize the theory. Our main interest here is in computing two-point functions of the stress tensor order-by-order in the loop counting parameter $G$.[6] There is some tension coming from two perspectives on this problem: on the one hand, the Nambu-Goto action with its square root is usually viewed as being problematic to quantize directly without ambiguity; on the other hand, the underlying theory is pure 3D gravity, which is expected to be renormalizable.

The subtlety in reconciling these perspectives has to do with the complicated (nonlinear and nonlocal) manner in which the symmetries of the gravitational description are realized once we pass to the reduced phase space description, and in particular with preserving these symmetries in the quantum theory. What we do concretely is compute the stress tensor correlators to two-loop order using dimensional regularization. At tree level and one-loop, the results are finite and unambiguous. At two-loops, we find that a single renormalization of the stress tensor is required and the divergent part comes as usual with an associated undetermined finite part parametrized here by $\mu$. For example, we find the $T_{zz} T_{zz}$ correlator at the two-loop order to be[7]

$$\langle T_{zz}(x) T_{zz}(0) \rangle = \frac{1}{z^4} \left[ \frac{c}{2} + 10(3 + 4G) \left( \frac{r_c}{z\overline{z}} \right)^2 + 96G \left( 8 + 60\ln(\mu^2 z\overline{z}) \right) \left( \frac{r_c}{z\overline{z}} \right)^3 + 2520G \left( \frac{r_c}{z\overline{z}} \right)^4 \right]. \quad (4)$$

Here $c = c_0 + 1 = \frac{3\ell_{\text{AdS}}}{2G} + 1$ is the 1-loop corrected Brown-Henneaux central charge [2,6] of the $r_c = 0$ theory. Regarding renormalizability, our result is therefore inconclusive: we suspect that the free parameter reflects that dimensional regularization is not preserving all symmetries, but further work is required to substantiate this, for example by imposing the relevant Ward identities.

Although we primarily focus on a flat planar boundary, it is also worthwhile to develop the curved boundary case. As preparation for this, we carefully work out the Chern-Simons

---

[4]Writing this in terms of $\phi = f + \overline{f}$ and $\pi_\phi = f' - \overline{f}'$ puts this in the more standard canonical form with kinetic term $\pi_\phi \dot{\phi}$.

[5]See (87) for all three components.

[6]Previous work [19,20] on this problem in the gravitational formulation stopped at tree level.

[7]The full set of two-point functions is written in (179).

formulation for general boundary metric. As an application, we show how to compute the action for Euclidean $AdS_3$ with finite $S^2$ boundary, including the large radius divergence associated with the Weyl anomaly of the boundary theory. This result is elementary to obtain in the metric description, but is somewhat subtle in the Chern-Simons formulation due to the need to introduce two overlapping patches for the gauge potentials.

We now mention some earlier related work. In previous work [19], a subset of us studied $AdS_3$ gravity with a finite cylinder boundary. One result was that the asymptotic Virasoro × Virasoro algebra was deformed in a precise and specific way by the breaking of conformal invariance associated with the finite boundary. Another result was that the free boundary graviton spectrum was deformed in the manner compatible with $T\overline{T}$ considerations. In the present paper, our main focus is on a planar boundary; this is simpler and we make other technical advances that allow us to go further than before. Stability and causality for gravity with cutoff boundary conditions is discussed in [21–23]. Covariant phase space in the presence of boundaries is reviewed in [24]. Jackiw-Teitelboim gravity [25, 26] at a finite boundary cutoff was studied in [27–29], with results relating to the spectrum of $T\overline{T}$-deformed quantum mechanics obtained in [30–32]. An important subtlety that arises, discussed in [27], is the distinction between microscopic versus effective theories of the JT gravity path integral, and the resulting nontrivial relations among the parameters and couplings; presumably these issues are also present in our context. Correlation functions in the 2D field theory or 3D bulk were studied in [20, 33–43]. Results in those papers were obtained either at low order in the $T\overline{T}$ coupling $\lambda_{T\overline{T}} \sim r_c/c$, or lowest order in the $1/c$ expansion (tree level in our language). An exception is [34] which proposed some all-orders results in $\lambda_{T\overline{T}}$. In the present work, our results hold to all orders in $r_c$ but are perturbative in $1/c$ (we go to two-loop order, extending the previous tree-level results). In the context of a massive scalar [36] and Dirac fermion [44], integrability was used to fix renormalization ambiguities. In [45–48], the $T\overline{T}$-deformed CS formulation of 3D gravity was discussed.

**Outline**

In section 2, we lay out some general principles involved in computing the action for boundary gravitons common to the metric and Chern-Simons formulations. In section 3 we discuss the metric formulation, developing a streamlined approach to computing the boundary action for a flat cutoff boundary. It is shown how to very easily obtain the Alekseev-Shatashvili action in the $r_c = 0$ limit. We also obtain the all orders action at finite $r_c$ in the special case of constant $(f', \overline{f}')$: the Nambu-Goto action. In section 4, we turn to the Chern-Simons formulation. Since it is of interest beyond the immediate concerns of this work, we carefully develop the variational principle for CS gravity with a general curved cutoff boundary. We carry out a perturbative computation of the action for gravitons on a cutoff planar boundary, obtaining results to eight order in $(f, \overline{f})$; the results turn out to match the expansion of the Nambu-Goto action to this order, leading us to conjecture that this extends to all orders. In section 5, we turn to correlation functions. We compute correlators of both elementary fields and the stress tensor. The 1-loop four-point function of elementary fields is found to require one counterterm in the action, and the 2-loop stress tensor correlators require a single renormalization of the stress tensor. We conclude with a brief discussion in section 6. Appendices give further details on the Chern-Simons formulation, including the comparison to the metric formulation, and a discussion of how to compute the action in the case of a spherical boundary. Another appendix gives details regarding the evaluation of Feynman diagrams.

## 2 Generalities on the phase space formulation of boundary graviton theories

In the next two sections, we obtain the action and stress tensor for boundary gravitons localized on a finite cutoff surface by working in the metric and Chern-Simons formulations respectively. These offer useful complementary perspectives and technical advantages, but the results of course agree. Here we discuss some general aspects of the problem to set the stage for the detailed analysis that follows.

The action (1), or its Chern-Simons equivalent, contains a mixture of "physical modes" and "pure gauge modes", and our goal is to arrive at a reduced action that omits the latter as much as possible. In general, one pays a price by reducing the degrees in the form of a loss of manifest symmetry, as for example in the light cone gauge treatment of Yang-Mills theory or string theory. In Yang-Mills perturbation theory, this price is typically too high and so a Lorentz invariant formulation with unphysical modes is usually adopted. However, in a topological theory, like pure 3D gravity, the reduction of degrees of freedom is so dramatic (removing all but the boundary modes) that the cost of losing some manifest symmetry is more than repaid.

We will construct a reduced action living on a flat boundary surface with coordinates $(x, t)$. The action is of the phase space variety, built out of a Hamiltonian $H$ and a "canonical 1-form" $\Upsilon$.[8] The phase space action in takes the form

$$I = -\int dt \left( i_{V_\eta} \Upsilon - H \right), \tag{5}$$

where $t$ is Euclidean time and $i_{V_\eta}$ denotes contraction with the phase space vector field $V_\eta$ that implements (Lorentzian) time translation. For example, for a particle moving in 1-dimension we might take $\Upsilon = p \delta q$ and $H(p, q) = \frac{p^2}{2m} + V(q)$. We have $i_{V_\eta} \Upsilon = -ip\dot{q}$ so that $I = \int dt \left( ip\dot{q} + H(p, q) \right)$.

The symplectic form $\Omega$ is given by $\Omega = \delta \Upsilon$. On the true phase space of the theory, $\Omega$ should be nondegenerate, meaning that $i_V \Omega = 0$ if and only if $V = 0$. In the context of gauge theory or gravity, it's natural to start with a larger "pre-phase space" with a degenerate, closed 2-form $\Omega$. The null directions of $\Omega$ on pre-phase space correspond to small gauge transformations. Part of our task here will be to remove the pure gauge modes corresponding to these null directions.

In the case of 3D gravity, the dynamical variables appearing in the phase space action will be fields $\left( f(x, t), \bar{f}(x, t) \right)$ on the boundary which therefore comprise the physical degrees of freedom.[9] The route to obtaining the action for these fields is a bit different in the metric versus Chern-Simons descriptions.

In metric formulation, the idea is to start with some reference solution and then apply boundary-condition-preserving coordinate transformations to construct a space of solutions. The symplectic form for gravity on pre-phase space was written down in [15], and implies that coordinate transformations that vanish at the boundary correspond to degenerate modes. All that matters is therefore the form of the coordinate transformation near the boundary, and this information is specified by the fields $(f, \bar{f})$. The coordinate transformations preserve the metric on the boundary, but change the value of the boundary stress tensor $T_{ij}$. As we discuss in detail in the next section, the phase space action follows immediately from the expressions for the boundary momentum density $p = \frac{i}{2\pi} T_{tx}$ and energy density $\mathcal{H} = \frac{1}{2\pi} T_{tt}$.

---

[8]Note that $\Upsilon$ is a 1-form on phase space, not on spacetime. Also, we will use $\delta$ to denote the exterior derivative on phase space, reserving $d$ for the exterior derivative on spacetime.

[9]More precisely, these fields are subject to residual gauge equivalences associated with isometries of $AdS_3$.

Turning to the Chern-Simons version, in this approach one can pass rather directly from the Chern-Simons action to the reduced phase space action once one has been sufficiently careful in defining boundary conditions and adding the associated boundary terms in the action. In the case of an asymptotic AdS$_3$ boundary, previous work on this problem includes [6, 49–51]. Our general approach follows [6], but since we work with a cutoff spacetime we first need to formulate a well defined variational principle and add the associated boundary terms to the action, since these differ from the ones used in most of the literature. We do this for a general curved boundary geometry, although our primary focus here is the case of a flat boundary. As usual in gauge theory, the time components of the gauge fields $A_t$ act as Lagrange multipliers imposing constraints [49]. Essentially, all one needs to do is to solve these constraint equations in a manner compatible with the boundary conditions, and then plug back into the action. The Lagrangian density is observed to be a total derivative, and the resulting boundary term is the desired phase space action. In this approach, the fields $(f, \overline{f})$ appear as free functions that parameterize solutions to the constraint equations and boundary conditions.

In either approach, obtaining the action (using perturbation theory if necessary) is rather mechanical, but the resulting expression may be unwieldy due to a suboptimal choice of coordinates on phase space. Especially for performing quantum mechanical perturbation theory, it is very convenient to choose coordinates such that the kinetic term in the action (the terms involving time derivatives) are purely quadratic in fields. This corresponds to choosing "Darboux coordinates" such that the components of the symplectic form are constant, which is always possible locally.[10] The $(f, \overline{f})$ are such Darboux coordinates, and part of our analysis will be to identify them. We will also find that it is possible[11] to choose these coordinates such that the Hamiltonian takes a simple form, namely that of the Nambu-Goto action (2). The Nambu-Goto action is well known to be the $T\overline{T}$-deformed action of a free scalar with canonical stress tensor [52]; the new features here are that the stress tensor derived from the gravity theory is not the canonical stress tensor, and the existence of a highly nontrivial field redefinition that relates the natural gravitational variables to those appearing in the Nambu-Goto action.

## 3 Metric formulation of boundary graviton action on the plane

### 3.1 Preliminaries

We start from the Euclidean signature action of 3D gravity with cosmological constant $\Lambda$,

$$I = -\frac{1}{16\pi G} \int_M d^3x \sqrt{g} \left( R - 2\Lambda \right) + I_{\text{bndy}} \,, \tag{6}$$

where the boundary terms are written below. Einstein's equations are then

$$R_{\mu\nu} - \frac{1}{2} R g_{\mu\nu} + \Lambda g_{\mu\nu} = 0 \,. \tag{7}$$

For $\Lambda < 0$, we define the AdS$_3$ radius as $\Lambda = -1/\ell_{\text{AdS}}^2$, and henceforth choose units such that $\ell_{\text{AdS}} = 1$. We impose Dirichlet boundary conditions on the metric of a two-dimensional boundary surface, and we choose coordinates $(r, x^i)$ such that this surface is located at $r = r_c$. The interior of the surface is taken to be the region $r > r_c$, so that the vector $\partial_r$ is inward pointing. It is convenient to choose Gaussian normal coordinates in the vicinity of the boundary

---

[10]In general coordinates, the complication is that the path integral measure is proportional to the nontrivial Pfaffian of the symplectic form.

[11]To at least eighth order in fields — we do not yet have a general proof.

so that the metric reads

$$ds^2 = \frac{dr^2}{4r^2} + g_{ij}(r,x)dx^i dx^j \,. \tag{8}$$

These coordinates may or may not break down away from the boundary, but this is largely immaterial for the purposes of studying the boundary graviton theory. The Dirichlet boundary condition means fixing $g_{ij}(r_c, x^i)$ as well as the form (8) (we could in principle fix only the induced metric on the boundary, but it is convenient to also put "gauge" conditions on the radial coordinate). The appropriate boundary action appearing in (6) is then

$$I_{\text{bndy}} = -\frac{1}{8\pi G}\int_{\partial M} d^2x\sqrt{\det g_{ij}}(K-1) - \frac{\log r_c}{32\pi G}\int d^2x\sqrt{\det g_{ij}}R(g_{ij})\,, \tag{9}$$

where the extrinsic curvature and its trace are

$$K_{ij} = -r\partial_r g_{ij}\,, \quad K = g^{ij}K_{ij}\,. \tag{10}$$

The terms in $I_{\text{bndy}}$ not involving $K$ depend only on the Dirichlet boundary data and so are not needed for a proper variational principle; however, they are added in order to ensure finiteness of the action in the asymptotic AdS$_3$ limit $r_c \to 0$.

The boundary stress tensor $T_{ij}$ is defined in terms of the on-shell variation of the action [53, 54],

$$\delta I = \frac{1}{4\pi}\int_{\partial M} d^2x\sqrt{\det g_{ij}}T^{ij}\delta g_{ij}\,, \tag{11}$$

and works out to be

$$T_{ij} = \frac{1}{4G}(K_{ij} - Kg_{ij} + g_{ij})\,. \tag{12}$$

In order to compare to a dual (deformed) CFT, we think of the latter as living on a rescaled metric $\gamma_{ij}$, defined as $\gamma_{ij} = r_c g_{ij}(r_c, x^i)$, which is in particular finite in the asymptotically AdS$_3$ case. The Einstein equations can be used to show that the stress tensor obeys the trace relation [33] [12]

$$\gamma^{ij}T_{ij} = \pi\lambda_{T\overline{T}}\det(\gamma^{ik}T_{kj}) - \frac{1}{8G}R(\gamma)\,, \tag{13}$$

with [9]

$$\lambda_{T\overline{T}} = \frac{4Gr_c}{\pi}\,. \tag{14}$$

On a flat surface, the $T\overline{T}$ deformation of the action is defined as

$$\partial_{\lambda_{T\overline{T}}}I_{\lambda_{T\overline{T}}} = -\frac{1}{4}\int d^2x\sqrt{\gamma}\det(\gamma^{ik}T_{kj})\,. \tag{15}$$

Using the definition of the stress tensor, this relation is implied by the trace relation (13). For our purposes, it will be more convenient to take the trace relation as the definition of the $T\overline{T}$ deformation.

---

[12]We emphasize (13) holds only for AdS$_3$. When $\Lambda > 0$, the trace relation, for example in dS$_3$, differs from (13) by an additional term $\propto \lambda_{T\overline{T}}^{-1}$. See [55–58] for discussions.

We now specialize to the case of a flat boundary metric,

$$g_{ij}(r_c)dx^i dx^j = \frac{1}{r_c}(dt^2 + dx^2) \,, \tag{16}$$

where $t$ lives on the real line and at this stage $x$ is allowed to live on either the line or the circle. Now, let $\xi^\mu \partial_\mu$ generate a diffeomorphism vector field that preserves the boundary conditions; namely, $(\nabla_\mu \xi_\nu + \nabla_\nu \xi_\mu)\big|_{r_c} = 0$. At this stage we should emphasize that we do not restrict to vector fields $\xi$ that are tangent to the boundary; we allow for $\xi$ with a nonzero normal component $\xi^r$, which in an active sense corresponds to moving the location of the boundary. To clarify this, note that a more geometrical characterization of our setup consists of finding flat surfaces embedded in an ambient AdS$_3$ background. To translate this picture into the one we actually use, we note that near each such surface we can construct a Gaussian normal coordinate system, with the surface at $r = r_c$ in these coordinates. The coordinate transformation needed to relate two such surfaces then clearly requires diffeomorphisms that are not tangent to the surfaces. Alternatively one could take the more algebraic view that the vector field $\xi$ is a computationally efficient way of representing a transformation on field space. Since the canonical formalism only cares that our transformations preserve the boundary conditions there is no difficulty posed by non-zero $\xi^r$. Associated to each boundary condition preserving vector field is a (not necessarily conserved) boundary charge [19,53][13]

$$Q[\xi] = \frac{i}{2\pi} \int T_{ti} \xi^i dx \,, \tag{17}$$

where the integral is evaluated on a constant $t$ slice of the boundary, and where the appearance of the $i$ is due to our choice of Euclidean signature. Translation invariance of the boundary metric implies the existence of conserved energy and momentum charges,

$$H = Q[-i\partial_t] = \frac{1}{2\pi} \int T_{tt} dx \,,$$

$$P = Q[\partial_x] = \frac{i}{2\pi} \int T_{tx} dx \,. \tag{18}$$

## 3.2 Phase space

Adopting the framework of covariant phase space [15,16], we think of phase space as the space of classical solutions that obey the boundary conditions (16) where we identify solutions related by "small gauge transformations," in a sense to be made precise momentarily. By definition, a phase space is equipped with a symplectic form $\Omega$, which is a non-degenerate, closed 2-form. For pure gravity in arbitrary dimension, a closed (and conserved) 2-form was written down in [15] as an integral over a Cauchy surface of an expression involving the metric and its derivatives. This object is degenerate on the space of all classical solutions, since it gives zero when contracted against an infinitesimal displacement in solution space corresponding to a coordinate transformation that vanishes at the boundary. To obtain the symplectic form we must therefore mod out by such coordinate transformations so that the 2-form becomes non-degenerate on the quotient space. In the context of AdS$_3$ gravity, this procedure can be made straightforward and concrete as follows.

We denote $\delta_\xi g_{\mu\nu}$ as the change of the metric under an infinitesimal coordinate transformation, $\delta_\xi g_{\mu\nu} = \nabla_\mu \xi_\mu + \nabla_\nu \xi_\mu$. We let $V_\xi$ denote the corresponding vector field on the space

---

[13]These charges are only conserved, and only generate symmetries, if $\xi$ is tangent to the boundary. It is then also a boundary Killing vector. We should also note that the notation $Q[\xi]$ is not to be confused with Wald's $Q_\xi$ [59] as explained in footnote 20 of [24].

of solutions; in terms of the Lie derivative, this corresponds to the statement $\delta_\xi g_{\mu\nu} = \mathcal{L}_{V_\xi} g_{\mu\nu}$. A key relation, verified by direct computation in [19], is[14]

$$i_{V_\xi} \Omega = -\delta Q[\xi] \, , \tag{19}$$

where $i$ denotes the contraction operation and $Q[\xi]$ is given by (17). Since $Q[\xi]$ takes the form of a boundary integral, this makes explicit the statement that diffeomorphisms which vanish at the boundary correspond to null directions of $\Omega$.

In order to give an explicit expression for the symplectic form, we need a correspondingly explicit description for the phase space (with small diffeomorphisms properly quotiented out). Since (19) states that diffeomorphisms that do not vanish at the boundary correspond to non-degenerate directions, it is natural to use such diffeomorphisms as our coordinates on phase space. So we start from some chosen reference solution and then perform all possible diffeomorphisms that preserve the boundary conditions. What this construction gives us is the "boundary graviton phase space" associated to the chosen reference solution. This is not necessarily the same as the full phase space, if in the latter one includes distinct solutions that cannot be related by a finite diffeomorphism. In the context of pure AdS$_3$ gravity we will take as our reference solution pure AdS$_3$ in either global or Poincaré coordinates, while candidates for distinct solutions are BTZ black holes and conical defects with different masses and angular momenta. However, BTZ black holes obey different boundary conditions than vacuum AdS$_3$ — i.e the BTZ black hole has two asymptotic boundaries in Lorentzian signature and a periodic time direction in Euclidean signature — while conical defects have singular metrics. In any event, what we are interested in studying here is the phase space of boundary gravitons living on the boundary of vacuum AdS$_3$.

We therefore start by writing down the reference metric which in the vicinity of the boundary takes the form (8). We then change coordinates, $r = r(r', x'^i)$, $x^i = x^i(r', x'^j)$ and demand that the metric at the boundary is unchanged,

$$ds^2\big|_{r'=r_c} = \frac{dr'^2}{4r_c^2} + g_{ij}(r_c, x'^i) dx'^i dx'^j \, . \tag{20}$$

That is, we demand that the metric components at the boundary takes the same form in the primed coordinates as they do in the original coordinates. Note, in particular, that in the new coordinates the location of the boundary is taken to be at $r' = r_c$; since this in general differs from $r = r_c$ we can think of the coordinate transformation as actively changing the location of the boundary. Imposing the boundary conditions in the new coordinates amounts to solving a system of PDEs for $r(r', x'^i)$ and $x^i(r', x'^j)$. Given the nature of the problem, it is natural to expand the coordinates transformation near the boundary, and so we write

$$\begin{aligned}
x &= x' + A(x', t') + (r' - r_c)U(x', t') + \dots \, , \\
t &= t' + B(x', t') + (r' - r_c)V(x', t') + \dots \, , \\
r &= r' + r'C(x', t') + r'^2 W(x', t') + \dots \, .
\end{aligned} \tag{21}$$

---

[14]Note that this result is valid even when $\xi$ has a component normal to the boundary, unlike what is often assumed, e.g. in [24]. This holds in pure gravity, where one can choose a gauge so that the quantity $C$ defined in [24] vanishes.

We then further expand around an initial time surface (taken to be $t = 0$) on the boundary,

$$x = x' + A(x') + \sum_{n=1}^{\infty} A_n(x') t'^n + (r' - r_c) \sum_{n=0}^{\infty} U_n(x') t'^n + \dots ,$$

$$t = t' + B(x') + \sum_{n=1}^{\infty} B_n(x') t'^n + (r' - r_c) \sum_{n=0}^{\infty} V_n(x') t'^n + \dots ,$$

$$r = r' + r' \sum_{n=0}^{\infty} C_n(x') t'^n + r'^2 \sum_{n=0}^{\infty} W_n(r') t'^n + \dots . \tag{22}$$

The reason for writing things in this way is that an inspection of the PDEs reveals that one can take as "initial data" any freely chosen functions $(A(x'), B(x'))$ (which respect periodicity conditions, if any, on $x$) and then determine the remaining functions in terms of these. In practice, at finite $r_c$ it seems difficult to solve this problem in closed form and so we will work perturbatively, treating the amplitudes of $(A, B)$ as small; this is equivalent to a small $G$ expansion. Perturbation theory is easy to carry out, since the functions $(U_n, V_n, C_n, W_n)$ are determined algebraically in terms of $(A, B)$ — no differential equations need to be solved — and one only needs a finite number of these functions at any given order.

The functions $(A(x'), B(x'))$ will, (modulo the gauge invariances to be discussed below) serve as coordinates on phase space. These functions determine the location of the boundary in the original reference spacetime. That is, the new boundary at $r' = r_c$ is located at $r = r_c + r_c C(x', t') + r_c^2 W(x', t') + \dots$, where $(C, W, \dots)$ are functions of $(A, B)$.

### 3.3 Boundary stress tensor

Going forward, the output of this analysis that we will need is an expression for the boundary stress tensor $T_{ij}$, evaluated at $t' = 0$, in terms of $(A, B)$. For this, we need the radial derivatives of the metric components evaluated at the boundary. It is straightforward to work this out to any desired order. Here we focus on the case that the reference solution is Poincaré AdS$_3$ given by (16) with $x$ noncompact. At quadratic order, the stress tensor works out to be

$$4GT_{tt} = -A''' + \frac{1}{2}\big((A'^2)'' + A''^2\big) - \frac{1}{2}\big((B'^2)'' + B''^2\big) - \frac{1}{2}(A''^2)'' r_c + \dots ,$$

$$4GT_{xt} = -B''' + (A'B')'' + A''B'' + (A'''B'')' r_c + \dots ,$$

$$4GT_{xx} = A''' - \frac{1}{2}\big((A'^2)'' + A''^2\big) + \frac{1}{2}\big((B'^2)'' + B''^2\big) + (A''A'''' - B''^2) r_c + \dots . \tag{23}$$

At higher orders, the expressions get more complicated but, as we discuss below, are greatly simplified after making an appropriate field redefinition.

The general structure of the stress tensor is fixed by gauge invariance (see section 3.6 for more detail). Consider the 4 parameter subgroup of the full 6 dimensional isometry group corresponding to translations, rotations, and dilatations of $(x, t)$ (with the dilatation accompanied by a rescaling of $r$). Since the stress tensor vanishes for the vacuum solution at $A = B = 0$, this means it must also vanish for any $(A = a + bx, B = c + dx)$ since any such coordinate transformation can be expressed as some combination of these isometries. Such choices of $(A, B)$ are therefore "pure gauge". From this we deduce that $(A, B)$ can only appear with a least one derivative, and that every term in the stress tensor must have at least one factor with at least two derivatives.

### 3.4 Symplectic form

We now discuss how to compute the symplectic form $\Omega$. We can use the relation $i_{V_\chi} \Omega = -\delta P$, where $\chi$ acts as an $x$-translation on the boundary, to efficiently deduce $\Omega$ given $P$. We first

note that the gauge invariance of the preceding paragraph implies that $P$ can always be put in the form, possibly after integrating by parts,

$$P = \int dx \Big( P_A(A,B) A' + P_B(A,B) B' \Big) , \tag{24}$$

where $P_{A,B}(A,B)$ are local functions of $(A,B)$ and their derivatives (regular as $A,B \to 0$), and are furthermore total derivatives of such local functions. The latter statement follows from the fact that each term in $P$ contains at least one factor with at least two derivatives. We now argue that the symplectic form is then given by $\Omega = \delta \Upsilon$ with

$$\Upsilon = \int dx \Big( P_A(A,B) \delta A + P_B(A,B) \delta B \Big) . \tag{25}$$

To compute $i_{V_\chi} \Omega$ and verify equality with $-\delta P$ we need expressions for $i_{V_\chi} \delta A$ and $i_{V_\chi} \delta B$. To this end, we start from (21) and perform a subsequent infinitesimal translation ($x' = x'' + \chi^x, t = t''$) and evaluate the effect at $t'' = 0$. For an $x$-translation, take $\chi = \chi^i \partial_i = \partial_x$. This acts as

$$A(x) \to A(x) + \chi^x + A'(x) \chi^x , \quad B(x) \to B(x) + B'(x) \chi^x , \tag{26}$$

so that

$$i_{V_\chi} \delta A = \chi^x + A'(x) \chi^x , \quad i_{V_\chi} \delta B = B'(x) \chi^x . \tag{27}$$

Using this to evaluate $i_{V_\chi} \Omega$, the first term in $i_{V_\chi} \delta A$ gives zero since $P_A$ is a total derivative. The remaining terms give

$$i_{V_\chi} \Omega = - \int dx \Big( P_A \delta A' + P_B \delta B' + \delta P_A A' + \delta P_B B' \Big) = -\delta P , \tag{28}$$

as desired.

We also need to establish that $\Omega$ defined above is the unique object obeying all requirements. Consider replacing $\Omega \to \Omega + \Delta\Omega$. We can always write $\Delta\Omega$ in the form

$$\Delta\Omega = \int dx \Big( \delta X_A(A,B) \wedge \delta A + \delta X_B(A,B) \wedge \delta B \Big), \tag{29}$$

with $X_{A,B}(A,B)$ being local functions with a perturbative expansion by the following argument. First, $\Delta\Omega$ is closed. Second, by gauge invariance each $A$ or $B$ in $X_{A,B}$ appears with at least one derivative, and $X_{A,B}$ must be total derivatives of local functions. Using these facts, we contract with the $x$-translation vector field and find

$$i_{V_\chi} \Delta\Omega = \delta \int dx \Big( X_A A' + X_B B' \Big) = -\delta \int dx \Big( X_A' A + X_B' B \Big) . \tag{30}$$

We now argue that this must vanish in order not to disturb the equality $i_{V_\chi} \Omega = -\delta P$. Using that $X_{A,B}$ admit a perturbative expansion and do not involve undifferentiated $A$ or $B$, it's easy to see that $i_{V_\chi} \Delta\Omega = 0$ requires $X_A' = X_B' = 0$. Since $X_A$ and $X_B$ are constants, they obey $\delta X_A = \delta X_B = 0$, which then implies $\Delta\Omega = 0$.

## 3.5 Action and equations of motion

From (21), the canonical equations of motion are

$$\dot{A} = A_1 , \quad \dot{B} = B_1 , \tag{31}$$

where we recall that $(A_1, B_1)$ are functions of $(A, B)$ and their derivatives. Related to this, if we consider a diffeomorphism by the time translation vector field $\eta^i \partial_i = \eta^t \partial_t$, we have

$$i_{V_\eta} \delta A = A_1 \eta^t , \quad i_{V_\eta} \delta B = \eta^t + B_1 \eta^t . \tag{32}$$

The equations of motion (31) are equivalent to the statement that $V_\eta$ is the Hamiltonian vector field corresponding to $Q[\eta]$, i.e. $i_{V_\eta} \Omega = -\delta Q[\eta]$, as this is a special case of (19). In particular, taking $\eta^t = -i$ gives $Q[\eta] = H$.

We now seek an action whose Euler-Lagrange equations coincide with these equations of motion, $i_{V_\eta} \Omega = -\delta H$. The answer is

$$I = -\int dt \left( i_{V_\eta} \Upsilon - H \right) = \int d^2 x \left( i P_A \dot{A} + i P_B \dot{B} + \mathcal{H} \right) , \tag{33}$$

where $\mathcal{H} = \frac{1}{2\pi} T_{tt}$ so that $H = \int dx \mathcal{H}$. The equality between the two lines uses the fact that $P_{A,B}$ are total derivatives so that the $\eta^t$ term in $i_{V_\eta} \delta B$ does not contribute. To see that this is the correct action, we evaluate $i_{V_\eta} \Omega$ as we did in (28) in the case of an $x$-translation. We now get

$$i_{V_\eta} \Omega = -i \int dx \left( \dot{P}_A \delta A - \delta P_A \dot{A} + \dot{P}_B \delta B - \delta P_B \dot{B} \right) = i \int dx \delta \left( P_A \dot{A} + P_B \dot{B} \right) , \tag{34}$$

which implies

$$i_{V_\eta} \Omega + \delta H = i \int dx \delta \left( P_A \dot{A} + P_B \dot{B} - i \mathcal{H} \right) . \tag{35}$$

It follows that the vanishing of $i_{V_\eta} \Omega + \delta H$ is equivalent to the Euler-Lagrange equations for (33). Using (23) the action at quadratic order is found to be

$$I = \frac{1}{16\pi G} \int d^2 x \left( A''' (\dot{B} - A') + B''' (\dot{A} + B') + \text{cubic} + \dots \right) . \tag{36}$$

The cubic and higher order terms are quite complicated when expressed in terms of $(A, B)$, but we will later find a field redefinition which greatly simplifies the action.

## 3.6 Symmetries

Our action (33) is invariant under certain gauge and global symmetries. We begin with the former, which originate due to isometries of the reference solution. AdS$_3$ has a six parameter group of coordinate transformations that leave the metric invariant, which in Lorentzian signature is SL$(2, \mathbb{R}) \times$ SL$(2, \mathbb{R})$. Applying one of these followed by a general coordinate transformation clearly has the same effect as applying only the latter, and this statement implies an equivalence relation between distinct $(A, B)$.

We'll adopt more compact notation, with $x^i = (x, t)$, writing (21) as

$$\begin{aligned}
x^i &= x'^i + A^i (x'^j) + (r' - r_c) U^i (x'^j) + \dots , \\
r &= r' + r' C (x'^j) + r'^2 W (x'^j) + \dots .
\end{aligned} \tag{37}$$

Let $\xi^\mu$ obey $\nabla^0_\mu \xi_\nu + \nabla^0_\nu \xi_\mu = 0$, where the derivatives are defined with respect to the reference solution $g^0_{\mu\nu}$. Writing the coordinate transformation $x^\mu = x'^\mu + \xi^\mu(x')$ in the form (37) defines $(A^i_\xi, U^i_\xi, C_\xi, F_\xi)$. We compose this with the subsequent transformation

$$
\begin{aligned}
x'^i &= x''^i + A^i(x''^i) + (r'' - r_c)U^i(x''^i) + \dots , \\
r' &= r'' + r'' C(x''^i) + r''^2 W(x''^i) + \dots ,
\end{aligned}
\tag{38}
$$

and evaluate at $(t'' = 0, r'' = r_c)$. This defines a transformation between $x^\mu$ and $x''^\mu$ labelled by some modified $A^i$ functions. This is equivalent to the transformation (37) in which $(x^i, r)$ appear on the left hand side. Explicitly, the gauge equivalence is

$$
A^i(x) \cong A^i(x) + \left[ A^i_\xi\big(x^j + A^j(x^k)\big) + \big(r_c C(x^j) + r_c^2 W(x^j)\big)U^i_\xi\big(x^j + A^j(x^k)\big)\right]\Big|_{t=0} .
\tag{39}
$$

Recalling that $C$ and $W$ are nonlinear functions of $A^i$ and their derivatives, we see that gauge transformations act in a complicated nonlinear way. The stress tensor is invariant under these transformations. The symplectic form is also invariant; this is not entirely obvious from our somewhat indirect method for extracting the symplectic form, but it is manifest when one expresses (as in [15]) the symplectic form as an integral over a spacelike surface, since that expression is expressed in terms of the metric, which is by definition invariant under the isometries. Furthermore, both the stress tensor and the symplectic form do not depend on time derivatives of $(A, B)$, so the invariance extends to transformations in which the parameters are allowed to have arbitrary dependence on time. This type of gauge symmetry was referred to as "quasi-local" in [6]. The phase space action is determined from the symplectic form and Hamiltonian, which implies that the action is also gauge invariant. As noted, the $SL(2, \mathbb{R}) \times SL(2, \mathbb{R})$ gauge transformations act on $(A, B)$ in a complicated nonlinear fashion. The formulas of course simplify markedly for $r_c = 0$, and we write out the corresponding transformations explicitly in the next section.

Global symmetries correspond to isometries of the metric on the boundary, which are simply translations and rotations (i.e. Poincaré transformations in real time). In this case, we first apply (37) followed by the infinitesimal transformation

$$
x'^i = x''^i + \epsilon^i + \theta^{ij} x''^j , \quad r' = r'' ,
\tag{40}
$$

with $\theta^{ij} = -\theta^{ji}$. Composing these transformations, we find

$$
\begin{aligned}
A(x) &\to A(x) + (1 + A')\epsilon^x + A_1 \epsilon^t + A_1 x \theta^{tx} , \\
B(x) &\to B(x) + (1 + B_1)\epsilon^t + B'\epsilon^x + B_1 x \theta^{tx} ,
\end{aligned}
\tag{41}
$$

where we used $\dot{A}|_{t=0} = A_1$ and $\dot{B}|_{t=0} = B_1$. These transformations are again highly nonlinear due to the appearance of $(A_1, B_1)$. The stress tensor transforms as a symmetric tensor under these translations and rotations.

## 3.7 Asymptotic AdS$_3$ case: Alekseev-Shatashvili action

For illustration, we consider the case of $r_c = 0$ where it is simple to carry out our general procedure in closed form. Starting from

$$
ds^2 = \frac{dr^2}{4r^2} + \frac{1}{r}dz d\bar{z} ,
\tag{42}
$$

the coordinate transformation [60]

$$z = F(z') - \frac{2r'F'^2\overline{F}''}{4F'\overline{F}' + r'F''\overline{F}''} \, ,$$

$$\overline{z} = \overline{F}(\overline{z}') - \frac{2r'\overline{F}'^2 F''}{4F'\overline{F}' + r'F''\overline{F}''} \, ,$$

$$r = \frac{16r'F'^3\overline{F}'^3}{(4F'\overline{F}' + r'F''\overline{F}'')^2} \, , \tag{43}$$

gives

$$ds^2 = \frac{dr'^2}{4r'^2} + \frac{1}{r'}dz'd\overline{z}' - \frac{1}{2}\{F, z'\}dz'^2 - \frac{1}{2}\{\overline{F}, \overline{z}'\}d\overline{z}'^2 + \frac{1}{4}r'\{F, z'\}\{\overline{F}, \overline{z}'\}dz'd\overline{z}' \, . \tag{44}$$

Here $F = F(z')$, $\overline{F} = \overline{F}(\overline{z}')$, primes on $(F, \overline{F})$ denote derivatives, and the Schwarzian derivative is

$$\{F, z'\} = \frac{F'''}{F'} - \frac{3}{2}\frac{F''^2}{F'^2} \, . \tag{45}$$

Writing $z = x + it$, and comparing (43) to (21) we read off

$$A + iB = F - z' \, ,$$

$$A - iB = \overline{F} - \overline{z}' \, ,$$

$$U + iV = -\frac{1}{2}\frac{F'\overline{F}''}{\overline{F}'} \, ,$$

$$U - iV = -\frac{1}{2}\frac{F''\overline{F}'}{F'} \, ,$$

$$C = F'\overline{F}' \, ,$$

$$W = -\frac{1}{4}F''\overline{F}'' \, . \tag{46}$$

The stress tensor is

$$T_{zz} = \frac{c_0}{12}\{F, z'\} \, , \quad T_{\overline{z}\overline{z}} = \frac{c_0}{12}\{\overline{F}, \overline{z}'\} \, , \quad T_{z\overline{z}} = 0 \, , \tag{47}$$

where $c_0 = \frac{3}{2G}$ is the Brown-Henneaux central charge. In this case, we of course could have written down (47) directly from knowledge of the asymptotic Virasoro symmetry of AdS$_3$ with $r_c = 0$, but here we are emulating the procedure we carry out for the general cutoff case. Expressing $T_{ij}$ in terms of $(A, B)$ gives the all orders version of the expressions (23) at $r_c = 0$.

From here, it is simple to work out the Alekseev-Shatashvili action for $(F, \overline{F})$. It's useful to generalize a bit by taking the stress tensor to be

$$T_{zz} = \frac{c_0}{12}\left(\frac{a}{2}F'^2 + \{F, z'\}\right) \, , \quad T_{\overline{z}\overline{z}} = \frac{c_0}{12}\left(\frac{a}{2}\overline{F}'^2 + \{\overline{F}, \overline{z}'\}\right) \, , \quad T_{z\overline{z}} = 0 \, , \tag{48}$$

where $a$ is a parameter that can be thought of as the stress tensor of a more general reference solution with $T_{zz} = T_{\overline{z}\overline{z}} = \frac{ac_0}{24}$. For example, $a = 1$ corresponds to global AdS$_3$ provided $x \cong x + 2\pi$, while values $0 < a < 1$ correspond to conical defect solutions. The energy and momentum from (18) are

$$H = -\frac{1}{2\pi}\int dx(T_{zz} + T_{\overline{z}\overline{z}}) = -\frac{c_0}{24\pi}\int dx\left(\frac{a}{2}F'^2 + \{F, z'\} + \frac{a}{2}\overline{F}'^2 + \{\overline{F}, \overline{z}'\}\right) \, ,$$

$$P = -\frac{1}{2\pi}\int dx(T_{zz} - T_{\overline{z}\overline{z}}) = -\frac{c_0}{24\pi}\int dx\left(\frac{a}{2}F'^2 + \{F, z'\} - \frac{a}{2}\overline{F}'^2 - \{\overline{F}, \overline{z}'\}\right) \, . \tag{49}$$

We now apply our general procedure to compute the symplectic form. This was previously stated in terms of $(A, B)$, but applies equally in terms of $(F, \overline{F})$. We are instructed to write $P$ in the form

$$P = \int dx \left( P_F F' + P_{\overline{F}} \overline{F}' \right) , \tag{50}$$

where $P_F$ and $P_{\overline{F}}$ are total derivatives. This is easily achieved using

$$\{F, z\} = -\frac{1}{2} \left( \frac{1}{F'} \right)'' F' + \text{total derivative} \tag{51}$$

yielding

$$P_F = -\frac{c_0}{48\pi} \left( a F' - \left( \frac{1}{F'} \right)'' \right) , \quad P_{\overline{F}} = \frac{c_0}{48\pi} \left( a \overline{F}' - \left( \frac{1}{\overline{F}'} \right)'' \right) . \tag{52}$$

The value of $\Upsilon$ (25) in this case gives

$$\Upsilon = -\frac{c_0}{48\pi} \int dx \left[ \left( a F' - \left( \frac{1}{F'} \right)'' \right) \delta F - \left( a \overline{F}' - \left( \frac{1}{\overline{F}'} \right)'' \right) \delta \overline{F} \right] . \tag{53}$$

The general formula (33) then yields the Alekseev-Shatashvili action [4]

$$I = -\frac{c_0}{24\pi} \int d^2x \left[ a F' \partial_{\overline{z}} F - \left( \frac{1}{F'} \right)'' \partial_{\overline{z}} F + a \overline{F}' \partial_z \overline{F} - \left( \frac{1}{\overline{F}'} \right)'' \partial_z \overline{F} \right] , \tag{54}$$

with

$$\partial_z = \frac{1}{2} (\partial_x - i \partial_t) , \quad \partial_{\overline{z}} = \frac{1}{2} (\partial_x + i \partial_t) . \tag{55}$$

The theory (54) describes a single (left and right moving) boson with variable central charge. As such, one expects it to be equivalent to the standard action for a free boson with a linear dilaton (or background charge) contribution to its stress tensor. Indeed, as noted in [4,5] the field redefinition

$$\left( e^{i\sqrt{a}F} \right)' = \sqrt{a} e^f , \quad \left( e^{-i\sqrt{a}\overline{F}} \right)' = \sqrt{a} e^{\overline{f}} \tag{56}$$

yields

$$T_{zz} = -\frac{c_0}{12} \left( \frac{1}{2} f'^2 - f'' \right) ,$$
$$T_{\overline{z}\overline{z}} = -\frac{c_0}{12} \left( \frac{1}{2} \overline{f}'^2 - \overline{f}'' \right) ,$$
$$\Upsilon = \frac{c_0}{48\pi} \int dx \left( f' \delta f - \overline{f}' \delta \overline{f} \right) ,$$
$$I = \frac{c_0}{96\pi} \int d^2x \left( f' \partial_{\overline{z}} f + \overline{f}' \partial_z \overline{f} \right) . \tag{57}$$

As $a \to 0$, the field redefinition reads

$$F' = e^f , \quad \overline{F}' = e^{\overline{f}} \tag{58}$$

and relations (57) continue to hold. Each chiral half of the action $I$ in (57) is the Floreanini-Jackiw action [61]; the two halves combined give a standard free scalar action in Hamiltonian

form. On the other hand, the stress tensor in (57) coming from gravity includes an improvement term (unlike what was considered in [46]). The improvement term is of course crucial, since without it the central charge would be fixed at $c = 1$.

Restricting now to $a = 0$, gauge transformations act as

$$\delta_\epsilon F = \sum_{n=0}^{2} \epsilon_n(t) F^n \,, \quad \delta_\epsilon \overline{F} = \sum_{n=0}^{2} \overline{\epsilon}_n(t) \overline{F}^n \,. \tag{59}$$

To compute the gauge variation of the stress tensor, symplectic form, and action we only need the transformations of $(f', \overline{f}')$ which are

$$\delta_\epsilon f' = 2\epsilon_2(t) e^f \,, \quad \delta_\epsilon \overline{f}' = 2\overline{\epsilon}_2(t) e^{\overline{f}} \,. \tag{60}$$

Gauge invariance of the stress tensor fixes the relative coefficient of the two terms in $T_{zz}$ and in $T_{\overline{z}\overline{z}}$. Using standard formulas from free boson CFT, it follows that correlators of stress tensors are those of a CFT with central charge $c = c_0 + 1$; we elaborate on this more in the course of our $r_c \neq 0$ discussion below.

It is also instructive to see how the stress tensor in (57) arises from Noether's theorem applied to the quadratic action in (57). An infinitesimal rigid translation of the boundary coordinates, $x^i \rightarrow x^i + \epsilon^i$ acts on $(F, \overline{F})$ as

$$\delta_\epsilon F = \partial_i F \epsilon^i \,, \quad \delta_\epsilon \overline{F} = \partial_i \overline{F} \epsilon^i \,, \tag{61}$$

arrived at by the same logic as led to (41). Actually, what we want is a transformation on phase space, and so we use the equations of motion $\partial_{\overline{z}} F = \partial_z \overline{F} = 0$ to trade away time derivatives and obtain

$$\delta_\epsilon F = F' \epsilon^z \,, \quad \delta_\epsilon = \overline{F}' \epsilon^{\overline{z}} \,. \tag{62}$$

As usual in the derivation of Noether's theorem, we now consider the transformation (62) in the case that $\epsilon^i$ depends arbitrarily on $x^i$. We then work out the transformation of $(f, \overline{f})$ as

$$\delta_\epsilon f = f' \epsilon^z + (\epsilon^z)' \,, \quad \delta_\epsilon \overline{f} = \overline{f}' \epsilon^{\overline{z}} + (\epsilon^{\overline{z}})' \,. \tag{63}$$

We finally compute the variation of the action and write it in the form

$$\delta I = -\frac{1}{2\pi} \int d^2 x \, T_{ij} \partial^i \epsilon^j \,, \tag{64}$$

yielding the stress tensor in (57).

## 3.8   Exact action for constant $(f', \overline{f}')$

Besides the asymptotic $r_c = 0$ limit, there is another special case in which we can derive the action to all orders. This is a consequence of the fact that we can find exact solutions of the boundary value problem in the case that second and higher derivatives of $f$ and $\overline{f}$ vanish. This leads to a result for the action which captures all dependence on $(f', \overline{f}')$, but not on higher derivatives of these fields.

We start from

$$ds^2 = \frac{d\rho^2}{4\rho^2} + \frac{1}{\rho} dw d\overline{w}, \tag{65}$$

and first perform the coordinate change

$$w = \frac{1-r}{1+r} e^{2\left(\frac{x}{1+r_c} + \frac{it}{1-r_c}\right)},$$
$$\overline{w} = \frac{1-r}{1+r} e^{2\left(\frac{x}{1+r_c} - \frac{it}{1-r_c}\right)}.$$
$$\rho = \frac{4r}{(1+r)^2} e^{\frac{4x}{1+r_c}}, \tag{66}$$

which gives the line element

$$ds^2 = \frac{dr^2}{4r^2} + \frac{1}{r}\left[\left(\frac{1-r}{1-r_c}\right)^2 dt^2 + \left(\frac{1+r}{1+r_c}\right)^2 dx^2\right]. \tag{67}$$

We now perform a further two-parameter coordinate redefinition that preserves the form of the metric at $r = r_c$. This corresponds to a rescaling by a parameter $a$,

$$r \to ar, \quad t \to \frac{\sqrt{a}(1-r_c)}{1-ar_c} t, \quad x \to \frac{\sqrt{a}(1+r_c)}{1+ar_c} x, \tag{68}$$

followed by a rotation by angle $\theta$,

$$t \to t\cos\theta + x\sin\theta, \quad x \to x\cos\theta - t\sin\theta. \tag{69}$$

Writing the combined transformation in the form $w = A(r)e^{f(x,t)}$ and $\overline{w} = \overline{A}(r)e^{\overline{f}(x,t)}$, we compute

$$f' = \frac{2\sqrt{a}}{1-a^2 r_c^2}(e^{i\theta} - ar_c e^{-i\theta}),$$
$$\overline{f}' = \frac{2\sqrt{a}}{1-a^2 r_c^2}(e^{-i\theta} - ar_c e^{i\theta}). \tag{70}$$

On the other hand, it is straightforward to compute the stress tensor in terms of $(a, \theta)$ and reexpress the results in terms of $(f', \overline{f}')$. Writing $P = \int dx\, p = \frac{i}{2\pi}\int T_{tx}dx$, we find

$$p = \frac{i}{2\pi} T_{tx} = \frac{1}{32\pi G}(f'^2 - \overline{f}'^2), \tag{71}$$

as in (79). Further, writing $H = \int dx\, h = \frac{1}{2\pi}\int T_{tt}dx$ we find that $h$ obeys

$$h - 4\pi G r_c(h^2 - p^2) = h_0, \tag{72}$$

where

$$h_0 = \frac{1}{32\pi G}(f'^2 + \overline{f}'^2). \tag{73}$$

Solving (72) for $h$ gives,

$$h = -\frac{1}{8\pi G r_c}\left(\sqrt{1 - 16\pi G r_c h_0 + (8\pi G r_c p)^2} - 1\right)$$
$$= -\frac{1}{8\pi G r_c}\left(\sqrt{1 - \frac{1}{2}r_c(f'^2 + \overline{f}'^2) + \frac{1}{16}r_c^2(f'^2 - \overline{f}'^2)^2} - 1\right), \tag{74}$$

where we chose the root which obeys $\lim_{r_c \to 0} h = h_0$.

This results implies that the full action takes the form

$$I = \frac{1}{32\pi G} \int d^2x \left[ if'\dot{f} - i\overline{f}'\dot{\overline{f}} - 4\frac{\sqrt{1 - \frac{1}{2}r_c(f'^2 + \overline{f}'^2) + \frac{1}{16}r_c^2(f'^2 - \overline{f}'^2)^2} - 1}{r_c} \right] \tag{75}$$
$$+ \text{ higher derivs },$$

where the higher derivative terms vanish when $f'' = 0$ and/or $\overline{f}'' = 0$. In the next section, we will find strong evidence that in fact the higher derivative terms are absent in general provided we define $(f, \overline{f})$ appropriately.

In this context, we can address the fact that the square root can become imaginary in some region of the $(f', \overline{f}')$ plane. From (70), we have

$$1 - \frac{1}{2}r_c(f'^2 + \overline{f}'^2) + \frac{1}{16}r_c^2(f'^2 - \overline{f}'^2)^2 = \left( \frac{1 - 2ar_c\cos(2\theta) + a^2r_c^2}{1 - a^2r_c^2} \right)^2 . \tag{76}$$

Therefore, the square root is real for any real value of $(a, \theta)$. It is natural to expect that in general (i.e. dropping the linearity assumption) the domain of $(f, \overline{f})$ is bounded such that the integrand in (75) is real, but this remains to be shown.

## 3.9 Boundary gravity action on planar cutoff

We now consider the general case, a planar boundary at $r = r_c$ with arbitrary functions $f$ and $\overline{f}$. As explained, the strategy is to start from the reference solution

$$ds^2 = \frac{dr^2}{4r^2} + \frac{dx^2 + dt^2}{r} , \tag{77}$$

and look for coordinate transformations, expressed in the form (21)-(22), such that

$$g_{r'r'}|_{r'=r_c} = \frac{1}{4r_c^2} , \quad g_{r'i'}|_{r'=r_c} = 0 , \quad g_{i'j'}|_{r'=r_c} = \frac{\delta_{ij}}{r_c} , \tag{78}$$

where $i$ and $j$ run over $(x, t)$. This problem can be solved order-by-order as an expansion in the freely specifiable functions $(A(x'), B(x'))$. Only algebraic equations need to be solved at each order, and the procedure is easily automated on the computer. What we need from this procedure are expressions for the components of the boundary stress tensor evaluated at $t' = 0$, which in turn depend on $\partial_{r'}g_{i'j'}|_{r'=r_c}$. The resulting expressions to quadratic order were written in (23).

At $r_c = 0$, we saw that the expressions for the stress tensor simplify dramatically under the field redefinition (58), and so we seek a version of this at nonzero $r_c$. As a criterion for what constitutes an optimal field redefinition, we note that in general the symplectic form $\Omega = \delta\Upsilon$ will have a complicated expansion in $(A, B)$. Quantization of the phase space action uses the natural measure $\text{Pf}(\Omega)$. A nontrivial measure is incorporated by expressing the Pfaffian as a fermionic path integral, but life is much simpler if the Pfaffian is constant, which is indeed the case at $r_c = 0$ after making the field redefinition. We therefore try to generalize this feature to nonzero $r_c$. Recall that the symplectic form is obtained from the momentum $P$ via the formulas (24)-(25).[15] We look to define new fields $(f, \overline{f})$ such that

$$P = \frac{i}{2\pi} \int T_{tx} dx = \frac{1}{32\pi G} \int dx(f'^2 - \overline{f}'^2), \tag{79}$$

---

[15] We argued for this using the $(A, B)$ fields, but we will see below that the argument also holds after the field redefinition to new fields $(f, \overline{f})$.

which implies

$$\Upsilon = \frac{1}{32\pi G}\int dx(f'\delta f - \overline{f}'\delta\overline{f})\,.\tag{80}$$

In particular, we want

$$T_{tx} = -\frac{i}{16G}(f'^2 - \overline{f}'^2) + (\text{total derivative})\,.\tag{81}$$

By explicit computation we find that this achieved by taking

$$A' + iB' = \exp\left[f - \frac{1}{4}r_c\overline{f}'^2 - \frac{1}{8}r_c^2\overline{f}'(f'\overline{f}')' - \frac{1}{16}r_c^2 f'(\overline{f}'^2)' + \dots\right] - 1\,,$$

$$A' - iB' = \exp\left[\overline{f} - \frac{1}{4}r_c f'^2 - \frac{1}{8}r_c^2 f'(f'\overline{f}')' - \frac{1}{16}r_c^2\overline{f}'(f'^2)' + \dots\right] - 1\,.\tag{82}$$

Actually, $T_{xt}$ takes the form (81) if we replace the 1/16 coefficient by anything; the value of 1/16 is chosen to simplify the form of the Hamiltonian, as noted below. This redefinition involves the spatial derivatives of $(A, B)$, so $(A, B)$ are nonlocally related to $(f, \overline{f})$. However, no undifferentiated $(A, B)$ will ever appear in the stress tensor, and so the stress tensor and action will be local in terms of $(f, \overline{f})$. The terms written in (82) are sufficient to work out the stress tensor and action up to quartic order in the new fields, which is sufficient for the two-loop computations we do in this work. For $T_{xt}$, we have

$$4GT_{xt} = -\frac{i}{4}(f'^2 - \overline{f}'^2) + \frac{i}{16}r_c^2\left(2f'f''\overline{f}' - f'^2\overline{f}''' - 2f''\overline{f}'\overline{f}'' + f'''\overline{f}'^2\right)'$$

$$+ \text{quartic tot. deriv.}\tag{83}$$

$T_{tt}$ is found to be

$$4GT_{tt} = \frac{1}{4}(f'^2 + \overline{f}'^2) - \frac{1}{16}r_c^2\left(f''\overline{f}'^2 + f'^2\overline{f}''\right)'' + \frac{1}{8}r_c f'^2\overline{f}'^2 + \text{quartic tot. deriv.}\tag{84}$$

The Hamiltonian to quartic order is thus

$$H = \int dx\,\mathcal{H} = \frac{1}{2\pi}\int dx\,T_{tt} = \frac{1}{16\pi G}\int dx\left(\frac{1}{2}f'^2 + \frac{1}{2}\overline{f}'^2 + \frac{1}{4}r_c f'^2\overline{f}'^2 + \dots\right)\,,\tag{85}$$

leading to the simple action

$$I = -\int dt\left(i_{V_\eta}\Upsilon - H\right) = \frac{1}{16\pi G}\int d^2x\left(f'\partial_{\overline{z}}f + \overline{f}'\partial_z\overline{f} + \frac{1}{4}r_c f'^2\overline{f}'^2 + \dots\right)\,,\tag{86}$$

where as usual $z = x + it$ and $\overline{z} = x - it$. This agrees with the expansion of (75) to this order, with no higher derivative terms present.[16] Coming back to our choice of the 1/16 in (82), for any other choice of coefficient the Hamiltonian includes a term proportional to $r_c^2 f'\overline{f}'f''\overline{f}''$. The full expressions for the stress tensor components to cubic order are

$$4GT_{zz} = \frac{1}{2}f'' - \frac{1}{4}f'^2 + \frac{1}{4}r_c f'''\overline{f}' - \frac{1}{8}r_c\left(f'^2 - 2f'\overline{f}'\right)'\overline{f}'$$

$$+ \frac{1}{16}r_c^2\left(f''''\overline{f}'^2 + (f'^2)''\overline{f}''\right) + \text{quartic}\,,$$

$$4GT_{\overline{z}\overline{z}} = \frac{1}{2}\overline{f}'' - \frac{1}{4}\overline{f}'^2 + \frac{1}{4}r_c\overline{f}'''f' - \frac{1}{8}r_c\left(\overline{f}'^2 - 2\overline{f}'f'\right)'f'$$

$$+ \frac{1}{16}r_c^2\left(\overline{f}''''f'^2 + (\overline{f}'^2)''f''\right) + \text{quartic}\,,$$

$$4GT_{z\overline{z}} = -\frac{1}{4}r_c f''\overline{f}'' + \frac{1}{8}r_c(f''\overline{f}'^2 + f'^2\overline{f}'') - \frac{1}{8}r_c^2(f'''\overline{f}'\overline{f}'' + f'f''\overline{f}''') + \text{quartic}\,.\tag{87}$$

---

[16]In the next section, we use the Chern-Simons formulation to verify (75) to eighth order.

It is easy to verify that stress tensor conservation follows from the Euler-Lagrange equations derived from (86), which is a good consistency check on our computations.

The action and stress tensor are invariant under the gauge transformation discussed in section 3.6. The complicated form of these transformations, along with the need to reexpress them in terms of $(f, \overline{f})$, make these symmetries difficult to use in practice. However, we expect that the full expressions for the stress tensor are fixed by gauge invariance in terms of their leading terms. Let us also comment that the stress tensor in principle can be derived via Noether's theorem using the transformations (41), as was done at $r_c = 0$ at the end of section 3.7.

Finally, we justify why we can pass from the momentum (79) to the canonical 1-form $\Upsilon$ in (80). As before, the question is whether the equation $i_{V_\chi} \Omega = -\delta P$ fixes the symplectic form $\Omega$ according to our rule, or whether there is an ambiguity of the form $\Delta \Omega = \int dx (\delta X_f \wedge \delta f + X_{\overline{f}} \wedge \delta \overline{f})$. To this end, we note that we can invert (82) order-by-order to obtain local expressions for $(f, \overline{f})$ in terms of $(A, B)$. Obviously, $(A, B)$ will always appear with a least one derivative. Given this, it follows that any candidate $\Delta \Omega$ of the form just noted will, under the field redefinition to $(A, B)$ turn into a $\Delta \Omega$ of the sort that we previously excluded. This justifies the procedure in the $(f, \overline{f})$ frame.

# 4 Chern-Simons formulation of boundary graviton action

Classically or in quantum perturbation theory, $2 + 1$-dimensional Einstein gravity can be formulated as a gauge theory, namely a Chern-Simons theory whose connections are constructed from the spin connection and vielbein of the first order formulation of general relativity. The relation between Chern-Simons theory and Einstein gravity at the non-perturbative level is unclear. One of the main ingredients in a non-perturbative theory of gravity is a sum over topologies, while such a sum is not natural from the perspective of gauge theory. These issues are however beyond the scope of this work. In this section, we are interested in understanding the perturbative theory of boundary gravitons from the Chern-Simons perspective.

The general strategy will be similar to section 3: in order to identify the boundary phase space, we consider all gauge transformations of a chosen solution that preserve the boundary conditions, and then quotient out small gauge transformations. Having identified the phase space, we evaluate the action. This will be done for the case of boundary conditions imposed at the asymptotic boundary of AdS$_3$, as well as the case of a finite cut-off boundary.

We start with a quick review of Chern-Simons gravity in three dimensions. In this section, we work in Lorentzian signature and only Wick rotate to Euclidean signature at the end of the computation to connect to the results of section 3.[17]

## 4.1 Action and boundary conditions

As mentioned above, Einstein gravity in 2+1 dimensions is classically equivalent to a gauge theory. For negative cosmological constant, the gauge group is $SO(2, 2) = SL(2, \mathbb{R}) \times SL(2, \mathbb{R})/\mathbb{Z}_2$. We denote the generators of $sl(2, \mathbb{R})$ by $L_{0, \pm 1}$ and take them to obey

$$[L_m, L_n] = (m - n) L_{m+n} . \tag{88}$$

An explicit representation is

$$L_0 = \begin{pmatrix} 1/2 & 0 \\ 0 & -1/2 \end{pmatrix}, \quad L_1 = \begin{pmatrix} 0 & 0 \\ -1 & 0 \end{pmatrix}, \quad L_{-1} = \begin{pmatrix} 0 & 1 \\ 0 & 0 \end{pmatrix}, \tag{89}$$

---

[17]To be precise, we will relate Lorentzian to Euclidean time by $t_L = i t_E$ and Lorentzian actions $S$ to Euclidean actions $I$ through $I = i S|_{t_L \to i t_E}$.

which obey

$$\mathrm{Tr}(L_0^2) = \frac{1}{2}\,, \quad \mathrm{Tr}(L_1 L_{-1}) = -1\,, \tag{90}$$

with other traces vanishing.

The Chern-Simons connections are related to the dreibein and the spin connection of the first order formulation of gravity as follows

$$A = L_a \left(\omega_\mu^a + e_\mu^a\right) dx^\mu\,, \quad \overline{A} = L_a \left(\omega_\mu^a - e_\mu^a\right) dx^\mu\,. \tag{91}$$

The base manifold $\mathcal{M}$ is equipped with coordinates $x^\mu = \{r, t, x\}$, where $r$ is the holographic coordinate for which $r \to 0$ at the conformal boundary. Greek indices will be reserved for the boundary $\partial \mathcal{M}$, which is equipped with coordinates $x^i = \{t, x\}$. The metric tensor can be extracted from the Chern-Simons connections as

$$g_{\mu\nu} = 2\,\mathrm{Tr}\left(e_\mu e_\nu\right) = \frac{1}{2}\,\mathrm{Tr}\left[(A-\overline{A})_\mu (A-\overline{A})_\nu\right]\,. \tag{92}$$

The gravitational action can be written in terms of the connections as

$$S_{\mathrm{grav}} = S_{\mathrm{CS}}[A] - S_{\mathrm{CS}}[\overline{A}] + S_{\mathrm{bndy}}\,, \tag{93}$$

where the Chern-Simons action at level $k$ reads

$$S_{\mathrm{CS}}[A] = \frac{k}{4\pi} \int \mathrm{Tr}\left(A \wedge dA + \frac{2}{3} A \wedge A \wedge A\right)\,. \tag{94}$$

Here $k$ is related to Newton's constant $G$ and the $\mathrm{AdS}_3$ length scale as[18]

$$k = \frac{\ell_{\mathrm{AdS}}}{4G}\,. \tag{95}$$

The equations of motion imply flatness of the Chern-Simons connections, which correspond to the Einstein equations and the vanishing of torsion.

We now turn to the choice of boundary conditions and the associated boundary term in the action. In complete generality, we write the connections as

$$\begin{aligned} A &= E^+ L_1 + \Omega L_0 + f^- L_{-1}\,, \\ \overline{A} &= f^+ L_1 + \overline{\Omega} L_0 + E^- L_{-1}\,, \end{aligned} \tag{96}$$

where at this stage all functions depend arbitrarily on all three coordinates. The corresponding metric is

$$ds^2 = \frac{1}{4}(\Omega - \overline{\Omega})^2 + (E^+ - f^+)(E^- - f^-)\,. \tag{97}$$

We choose boundary conditions that mimic our construction in the metric formulation, where we choose to fix all metric components at $r = r_c$. We therefore write

$$\left(\Omega - \overline{\Omega}\right)\big|_{r_c} = \frac{1}{r_c} dr\,, \quad (E - f)^\pm\big|_{r_c} = \pm 2 e_i^\pm dx^i\,, \tag{98}$$

so that

$$ds^2\big|_{r_c} = \frac{dr^2}{4r_c^2} - 4 e_i^+ e_j^- dx^i dx^j\,. \tag{99}$$

---

[18] We are again working with $\ell_{\mathrm{AdS}} = 1$ in this section.

Here, $e_i^\pm$ is the fixed boundary zweibein. A boundary term compatible with these boundary conditions is [19]

$$S_{\text{bndy}} = \frac{k}{4\pi} \int_{\partial\mathcal{M}} \text{Tr} A \wedge \overline{A} - \frac{k}{4\pi} \int_{\partial\mathcal{M}} \text{Tr}\left[ L_0(A - \overline{A}) \wedge (A - \overline{A}) \right]. \tag{100}$$

In particular, a straightforward computation yields the following on-shell variation of the action

$$\delta S_{\text{bulk}} + \delta S_{\text{bndy}} = \frac{k}{2\pi} \int_{\partial M} \left( f^- \wedge \delta e^+ + f^+ \wedge \delta e^- \right). \tag{101}$$

Since only variations of the fixed quantities $e^\pm$ appear, our variational principle is consistent.

In practice, it is convenient to impose additional boundary conditions which are compatible with the variational principle and incorporate all solutions of interest. The boundary spin connection $\omega$, given by

$$\frac{1}{2}\left(\Omega + \overline{\Omega}\right)\big|_{r_c} = \omega, \tag{102}$$

is so far unfixed. However, vanishing of the Chern-Simons field strength implies

$$de^+ - \omega \wedge e^+ = de^- + \omega \wedge e^- = 0, \tag{103}$$

which are the usual torsion-less conditions that uniquely fix the boundary spin connection $\omega$ in terms of the vielbein $e^\pm$. We therefore impose (103) as a boundary condition. The remaining flatness conditions evaluated at the boundary impose conservation of the stress tensor (defined below) and also fix its trace.

## 4.2 Stress tensor

The boundary stress tensor is defined in terms of the on-shell variation of the action as[20]

$$\delta S = \frac{1}{4\pi} \int d^2x \sqrt{g}\, T^{ij} \delta g_{ij} = \frac{1}{\pi} \int d^2x \det e\, T_a^i \delta e_i^a. \tag{104}$$

Comparing to (101), we read off

$$T_+^j = k\epsilon^{ij} f_i^-, \quad T_-^j = k\epsilon^{ij} f_i^+, \tag{105}$$

where our boundary orientation is defined by

$$\epsilon^{tx} = -\epsilon^{xt} = \frac{1}{\det e}. \tag{106}$$

## 4.3 Relation to metric formulation

It is helpful to write the relation between the bulk and boundary terms in the metric versus Chern-Simons descriptions. For the bulk Einstein-Hilbert action, we have

$$\frac{1}{16\pi G} \int_M \sqrt{g}(R + 2) = S_{\text{CS}}[A] - S_{\text{CS}}[\overline{A}] - \frac{k}{4\pi} \int_M d\,\text{Tr}(A \wedge \overline{A}). \tag{107}$$

---

[19]This boundary term appears in a different form in [45].

[20]In this formula $\sqrt{g}$ and $\det e$ refer to boundary quantities, related by $\sqrt{g} = 2\det e$ according to our convention (91).

For the Gibbons-Hawking terms, where $h_{\mu\nu}$ is the boundary metric, we explain in appendix D how it can be rewritten as

$$\frac{1}{8\pi G}\int_{\partial M} d^2x \sqrt{h}K = \frac{k}{2\pi}\int_{\partial M} \text{Tr}(A\wedge\overline{A})\,, \tag{108}$$

under the condition $\partial_a n^a = 0$, which is satisfied given our choice of gauge (98). We finally have the boundary area counterterm,

$$-\frac{1}{8\pi G}\int_{\partial M} \sqrt{h} = -\frac{k}{4\pi}\int_{\partial M}\text{Tr}[L_0(A-\overline{A})\wedge(A-\overline{A})]\,. \tag{109}$$

The relationship between the complete actions is therefore[21]

$$\frac{1}{16\pi G}\int_M \sqrt{g}(R+2) + \frac{1}{8\pi G}\int_{\partial M} d^2x\sqrt{h}K - \frac{1}{8\pi G}\int_{\partial M}\sqrt{h}$$

$$= S_{\text{CS}}[A] - S_{\text{CS}}[\overline{A}] + \frac{k}{4\pi}\int_{\partial M}\text{Tr}(A\wedge\overline{A}) - \frac{k}{4\pi}\int_{\partial M}\text{Tr}[L_0(A-\overline{A})\wedge(A-\overline{A})]. \tag{110}$$

So our Chern-Simons action agrees with the standard gravity action.

## 4.4 Boundary action

We now reduce the bulk theory to the boundary by solving the constraints of the theory and plugging back in. The Chern-Simons connections can be written in a space-time split as

$$A = A_t dt + \tilde{A},\quad \overline{A} = \overline{A}_t dt + \tilde{\overline{A}}, \tag{111}$$

and we similarly write the exterior derivative on spacetime as $d = dt\,\partial_t + \tilde{d}$. The components $A_t$ and $\overline{A}_t$ appear in the action as Lagrange multipliers,

$$S_{CS}[A] = \frac{k}{4\pi}\int_{\mathcal{M}}\text{Tr}\left[2A_t dt \wedge\tilde{\mathcal{F}} + \tilde{A}\wedge dt\wedge\partial_t\tilde{A} - \tilde{d}\left(\tilde{A}\wedge A_t dt\right)\right]\,, \tag{112}$$

where the spatial field strength is $\tilde{\mathcal{F}} = \tilde{d}\tilde{A} + \tilde{A}\wedge\tilde{A}$. The $A_t$ equation imposes the constraint that the spatial components of the field strength (and its barred counterpart) must vanish,

$$\tilde{\mathcal{F}} = \tilde{\overline{\mathcal{F}}} = 0\,. \tag{113}$$

These constraints are solved by writing

$$\tilde{A} = g^{-1}\tilde{d}g\,,\quad \tilde{\overline{A}} = \overline{g}^{-1}\tilde{d}\overline{g}\,. \tag{114}$$

We write the group elements in a Gauss parametrization

$$g = \begin{pmatrix} 1 & 0 \\ -F & 1 \end{pmatrix}\begin{pmatrix} \lambda & 0 \\ 0 & \lambda^{-1} \end{pmatrix}\begin{pmatrix} 1 & \Psi \\ 0 & 1 \end{pmatrix},$$

$$\overline{g} = \begin{pmatrix} 1 & \overline{F} \\ 0 & 1 \end{pmatrix}\begin{pmatrix} \overline{\lambda}^{-1} & 0 \\ 0 & \overline{\lambda} \end{pmatrix}\begin{pmatrix} 1 & 0 \\ -\overline{\Psi} & 1 \end{pmatrix}. \tag{115}$$

---

[21]Note that we are defining the action with an overall sign flip compared (6); this has to do with the fact that (6) was defined in Euclidean signature.

It is a straightforward exercise to rewrite the boundary conditions (98) in terms of the functions appearing in (115). The $(\Psi, \overline{\Psi})$ are determined as

$$\Psi = -\frac{\lambda'}{\lambda^3 F'} + \frac{\omega_x}{2\lambda^2 F'} \,,$$

$$\overline{\Psi} = -\frac{\overline{\lambda}'}{\overline{\lambda}^3 \overline{F}'} - \frac{\omega_x}{2\overline{\lambda}^2 \overline{F}'} \,, \tag{116}$$

where $\omega_x$ is the space component of the boundary spin connection, fixed in terms of the boundary vielbein. The remaining boundary conditions amount to the following differential equations

$$2e_x^+ = \lambda^2 F' - \overline{\Psi}' - \overline{\lambda}^2 \overline{\Psi}^2 \overline{F}' - \frac{2\overline{\Psi}}{\overline{\lambda}} \overline{\lambda}' \,,$$

$$-2e_x^- = \overline{\lambda}^2 \overline{F}' - \Psi' - \lambda^2 \Psi^2 F' - \frac{2\Psi}{\lambda} \lambda' \,. \tag{117}$$

The equations (116) and (117) are to be imposed at the boundary surface $r = r_c$.

Having chosen the Gauss parametrization, one finds that the bulk Lagrangian becomes a total derivative, and so the complete action takes the form of a boundary term. After some algebra (see appendix B), this can be written as

$$S_{\text{grav}} = -\frac{k}{2\pi} \int d^2x \left( \frac{\lambda' \partial_t \lambda}{\lambda^2} - \lambda^2 F' \partial_t \Psi \right) - \frac{k}{\pi} \int d^2x \left( \lambda^2 \Psi^2 F' + \Psi' + \frac{2\Psi \lambda'}{\lambda} \right) e_t^+$$

$$+ \frac{k}{2\pi} \int d^2x \left( \frac{\overline{\lambda}' \partial_t \overline{\lambda}}{\overline{\lambda}^2} - \overline{\lambda}^2 \overline{F}' \partial_t \overline{\Psi} \right) - \frac{k}{\pi} \int d^2x \left( \overline{\lambda}^2 \overline{\Psi}^2 \overline{F}' + \overline{\Psi}' + \frac{2\overline{\Psi} \overline{\lambda}'}{\overline{\lambda}} \right) e_t^- \,. \tag{118}$$

The boundary conditions (116)-(117) imply four equations for the six Gauss functions, leaving two free functions, which we can take to be $(F, \overline{F})$. So, in principle, we should use (116) and (117) to obtain $(\Psi, \overline{\Psi}, \lambda, \overline{\lambda})$ in terms of $(F, \overline{F})$ and plug into (118) to obtain the reduced action. However, in practice it is not possible to carry this out analytically[22]. To obtain explicit results we either need to consider the asymptotic AdS$_3$ case of $r_c \to 0$, or use perturbation theory. We discuss these in turn below.

One feature to keep in mind is that we only need to solve for the Gauss functions on the cutoff surface. These functions determine the connections restricted to that surface, which we call $(a, \overline{a})$. The full connections $(A, \overline{A})$ may then be determined away from the boundary by the construction

$$A = b^{-1} a b + b^{-1} db \,, \quad \overline{A} = b \overline{a} b^{-1} + b db^{-1} \,, \tag{119}$$

where

$$b = e^{-\frac{1}{2} \ln\left(\frac{r}{r_c}\right) L_0} \,, \tag{120}$$

and with $a$ and $\overline{a}$ functions of only the boundary coordinates. This is the Chern-Simons equivalent of radial gauge, which we can always choose at least in a neighborhood of the boundary. Flat boundary connections $(a, \overline{a})$ are thereby promoted to flat bulk connections $(A, \overline{A})$.

---

[22]Even though it is not possible to solve the boundary conditions analytically, they do have a beautiful physical interpretation. They correspond to the definition of the stress tensor in a $T\overline{T}$-deformed theory, understood as a theory coupled to topological gravity. See appendix C for more details.

### 4.5 Asymptotic boundary ($r_c \to 0$)

In this subsection, we consider imposing boundary conditions at the asymptotic boundary of AdS$_3$. The results obtained here can be found in [6].

Asymptotically AdS$_3$ boundary conditions correspond to taking $r_c \to 0$ with boundary vielbein $e_\mu^a \sim r_c^{-1/2}$. The boundary conditions (116) and (117) imply $(\Psi, \overline{\Psi}) \sim r_c^{1/2}$ and $(\lambda, \overline{\lambda}) \sim r_c^{-1/4}$, while $(F, \overline{F})$ stay finite. The solution for (117) reads

$$\lambda = \sqrt{\frac{2e_x^+}{F'}}, \quad \overline{\lambda} = \sqrt{\frac{-2e_x^-}{\overline{F}'}}, \tag{121}$$

while the boundary action evaluates to

$$S_{\text{grav}} = -\frac{k}{\pi} \int d^2x \left( \frac{\lambda' D\lambda}{\lambda^2} - \lambda^2 F' D\Psi \right) + \frac{k}{\pi} \int d^2x \left( \frac{\overline{\lambda}' \overline{D\lambda}}{\overline{\lambda}^2} - \overline{\lambda}^2 \overline{F}' \overline{D\Psi} \right)$$
$$- \frac{k}{8\pi} \int d^2x \frac{e_t^+ e_x^- - e_t^- e_x^+}{e_x^+ e_x^-} \omega_x^2, \tag{122}$$

with

$$D = \frac{1}{2} \left( \partial_t - \frac{e_t^+}{e_x^+} \partial_x \right), \quad \text{and} \quad \overline{D} = \frac{1}{2} \left( \partial_t - \frac{e_t^-}{e_x^-} \partial_x \right). \tag{123}$$

The term in the second line of (122) is a constant determined by the boundary conditions. For a flat planar boundary, we arrive at the Alekseev-Shatashvili action

$$S_{\text{grav}} = S_{\text{AS}}[F] + S_{\text{AS}}[\overline{F}], \tag{124}$$

with

$$S_{\text{AS}}[F] = \frac{k}{4\pi} \int d^2x \left( \frac{1}{F'} \right)'' \partial_{\overline{z}} F$$
$$= \frac{k}{4\pi} \int_{\partial \mathcal{M}} \left[ \frac{\dot{F}}{F'} \left( \frac{F'''}{F'} - \frac{F''^2}{2F'^2} \right) + \frac{F'''}{F'} - \frac{3}{2} \frac{F''^2}{F'^2} \right]. \tag{125}$$

As noted previously by (58), the field redefinition

$$F' = e^f, \quad \overline{F}' = e^{\overline{f}}, \tag{126}$$

yields the free boson action

$$S_{\text{grav}}[f, \overline{f}] = -\frac{k}{4\pi} \int d^2x \left[ f' \partial_z f + \overline{f}' \partial_{\overline{z}} \overline{f} \right]. \tag{127}$$

### 4.6 Perturbation theory for planar cutoff boundary

We now consider the case of a boundary at a finite cut-off $r = r_c$, with the simplifying assumption of a flat boundary geometry. We will be able to solve the boundary conditions (117) by perturbing around a reference solution. Explicitly, we keep $r_c$ finite and fixed, and take the boundary vielbein corresponding to a flat plane

$$e^+ = \frac{1}{2\sqrt{r_c}}(dx + dt), \quad e^- = -\frac{1}{2\sqrt{r_c}}(dx - dt). \tag{128}$$

The corresponding solution to (103) is $\omega_x = 0$. We will perturb around the solution

$$F^{(0)} = \overline{F}^{(0)} = x \,, \tag{129}$$

which implies $\Psi^{(0)} = \overline{\Psi}^{(0)} = 0$ and $\lambda^{(0)} = \overline{\lambda}^{(0)} = r_c^{-1/4}$. This solution corresponds to the background metric $ds^2 = \frac{dr^2}{4r^2} + \frac{1}{r}(-dt^2 + dx^2)$, i.e. Poincaré AdS$_3$.

Having identified a background field configuration, we expand around it order-by-order. We adapt the following notation for the perturbations:

$$\begin{aligned}
\lambda^2 &= \frac{1}{\sqrt{r_c}} \left( 1 + f + f^{(2)} + f^{(3)} + \cdots \right), \\
\overline{\lambda}^2 &= \frac{1}{\sqrt{r_c}} \left( 1 + \overline{f} + \overline{f}^{(2)} + \overline{f}^{(3)} + \cdots \right), \\
F' &= 1 + g^{(1)} + g^{(2)} + g^{(3)} + \cdots, \\
\overline{F}' &= 1 + \overline{g}^{(1)} + \overline{g}^{(2)} + \overline{g}^{(3)} + \cdots.
\end{aligned} \tag{130}$$

We will regard $f$ and $\overline{f}$ as the fundamental fields of our perturbative action, while $f^{(i)}$, $g^{(i)}$ and their barred counter-parts will be chosen so that the boundary conditions (117) are satisfied perturbatively. The boundary conditions fully determine $g^{(i)}$ while the functions $f^{(i)}$ can be chosen freely. This freedom amounts to a field redefinition of $(f, \overline{f})$ which will be used to obtain the simplest action possible.

Solving the boundary conditions perturbatively, which means working order-by-order in the amplitudes of $(f, \overline{f})$, we find the following expressions for the first few functions $g^{(i)}$:

$$\begin{aligned}
g^{(1)} &= -f - \frac{r_c}{2} \overline{f}'', \\
g^{(2)} &= f^2 - f^{(2)} + \frac{r_c}{4} \left( \overline{f}'^2 + 2f\overline{f}'' + 2\overline{f}f'' - 2f^{(2)''} \right) - \frac{1}{r_c^2} \left( f''\overline{f}'' + \overline{f}'f''' \right),
\end{aligned} \tag{131}$$

and similarly for $\overline{g}^{(i)}$. The formulas for higher order terms are easily found since the boundary conditions amount to linear equations for $g^{(i)}$. However, their expressions are not illuminating and get messy at higher orders, so we do not write them explicitly.

As mentioned above, we are free to choose the functions $f^{(i)}$, which amounts to a choice of field redefinition. Just as we found in the metric formulation, a judicious choice simplifies the expression for the action greatly. We first of all demand

$$4GT_{xt} = \frac{1}{4} \left( \overline{f}'^2 - f'^2 \right) + \text{total derivatives} \,, \tag{132}$$

which implies a simple expression for the part of the action involving time derivatives, agreeing with (127), and essentially corresponds to choosing Darboux coordinates. The field redefinition that achieves this reads

$$\begin{aligned}
f^{(2)} &= \frac{1}{2} f^2 - \frac{r_c}{2} f'\overline{f}' + \dots, \\
f^{(3)} &= \frac{1}{6} f^3 - \frac{r_c}{2} f f'\overline{f}' + \frac{r_c^2}{4} \left[ \frac{1}{2} \overline{f}'^2 f'' + f'^2 \overline{f}'' + \dots \right],
\end{aligned} \tag{133}$$

and similarly for the barred functions. The second condition that can be satisfied is that all higher derivatives of $f$ and $\overline{f}$ can be canceled in the action, i.e. only powers of first derivatives

appear. This condition first appears at fourth order, where the appropriate choice of field redefinition reads

$$f^{(4)} = \frac{1}{4!}f^4 - \frac{r_c}{4}f^2 f'\overline{f}' - \frac{r_c^2}{8}(f'^3\overline{f}' - ff''\overline{f}'^2 - 2ff'^2\overline{f}'' + f'\overline{f}'^3 - 2f'^2\overline{f}'^2)$$
$$- \frac{r_c^3}{16}(4f'f''\overline{f}'\overline{f}'' + \tfrac{1}{3}f'''\overline{f}'^3 + f'^3\overline{f}''') \,. \tag{134}$$

Perturbation theory subject to these conditions can be automated using computer algebra software (we used Mathematica) and performed to higher orders. One useful observation is that the terms needed in the choice of $f^{(n)}$ and $\overline{f}^{(n)}$ to satisfy the two aforementioned conditions already appear (with different coefficients) in the Hamiltonian density at order $n-1$. More specifically, the Hamiltonian density has a simple expression up to a total double derivative contribution, and the terms that appear in this double derivative are exactly the ones that make up our choice of $f^{(n)}$ and $\overline{f}^{(n)}$.

We carried out this perturbation theory to the 8th order (i.e. computing all terms of the schematic form $f^m\overline{f}^m$ with $n+m \leq 8$). The result coincides with the expansion to this order of the Nambu-Goto action (2). We naturally conjecture that this result extends to all orders, but we do not have a proof.

This analysis also yields expressions for the boundary stress tensor $T_{ij}$ to 8th order. These agree with the expressions found in the metric formulation (up to 4th order, which is as far as we pushed the computation in the metric formulation). Since our computations below only use the stress tensor up to cubic order, written in (87), we refrain from writing the higher order expressions, which rapidly become complicated.

# 5 Correlation functions

In this section, we discuss the computation of correlation functions of the fundamental fields $(f, \overline{f})$ and the stress tensor $T_{ij}$. We will work up to two-loop order where, as seen from (86), $G$ acts as a loop counting parameter.

There are some subtleties having to do with the realization of symmetries in this theory. For example, the action is not manifestly Lorentz invariant, even though the underlying theory is Lorentz invariant since it was obtained by expanding around a Lorentz invariant background (the flat plane). We expect that the stress tensor should behave in correlators like a Lorentz tensor. As was discussed above, Lorentz symmetry is realized nonlinearly on the $(f, \overline{f})$ fields. A general phenomenon that can occur when doing perturbation theory in a QFT with a nonlinearly realized symmetry is that one encounters divergent terms that are not invariant under the symmetry. One then needs to perform a field redefinition to restore the symmetry (or equivalently, to modify the symmetry transformation), e.g. [62]. Another approach is to modify the theory off-shell so as to preserve the symmetry, e.g. [63]. Our approach is to modify perturbation theory in a way that maintains Lorentz invariance while only changing contact terms in correlators. In particular, correlation functions of stress tensors at non-coincident points will respect Lorentz invariance.

## 5.1 Action

We found that the action to quartic order is

$$I = \frac{1}{16\pi G} \int d^2x \left( f'\partial_{\overline{z}}f + \overline{f}'\partial_z\overline{f} + \frac{1}{4}r_c f'^2\overline{f}'^2 + \dots \right) \,. \tag{135}$$

Recall that $z = x + it$ and $\bar{z} = x - it$ so that

$$\partial_z = \frac{1}{2}(\partial_x - i\partial_t), \quad \partial_{\bar{z}} = \frac{1}{2}(\partial_x + i\partial_t). \tag{136}$$

Here $G$ is the loop counting parameter. In particular, since the stress tensor also has a $1/G$ prefactor, it follows that an $L$ loop contribution to a stress tensor correlator has dependence $G^{L-1}$.

## 5.2 Propagator

Let's first discuss the propagators in momentum space using the Fourier transform convention

$$\psi(x,t) = \int \frac{d^2p}{(2\pi)^2} \psi(p) e^{ip_t t + ip_x x}, \tag{137}$$

or in complex coordinates

$$\psi(z,\bar{z}) = \int \frac{d^2p}{(2\pi)^2} \psi(p) e^{ip_z z + ip_{\bar{z}} \bar{z}}, \tag{138}$$

with

$$p_x = p_z + p_{\bar{z}}, \quad p_t = i(p_z - p_{\bar{z}}). \tag{139}$$

Note also that

$$p^2 = p_t^2 + p_x^2 = 4p_z p_{\bar{z}}. \tag{140}$$

The free 2-point functions are then

$$\langle f'(p)f'(p')\rangle_0 = 32\pi G \frac{p_x p_z}{p^2}(2\pi)^2 \delta^2(p+p'), \quad \langle \overline{f}'(p)\overline{f}'(p')\rangle_0 = 32\pi G \frac{p_x p_{\bar{z}}}{p^2}(2\pi)^2 \delta^2(p+p'). \tag{141}$$

We wrote the results for the fields with an $x$-derivative since $(f, \overline{f})$ always appear in the action and stress tensor with at least one $x$-derivative.

We will be using dimensional regularization to compute loop diagrams. Our convention for going from 2 to $d$ dimensions is that we introduce $d - 2$ new spatial dimensions. We continue to refer to momenta in the original two dimensions by $(p_x, p_t)$ or $(p_z, p_{\bar{z}})$, but $p^2$ is taken to run over all dimensions: $p^2 = p_t^2 + p_x^2 + \sum_{i=2}^d p_i^2$. In particular, the relation (140) only holds in $d = 2$.

Coming back to the propagators, after stripping off delta functions and using (139), we have

$$\langle f'(p)f'(-p)\rangle_0 = 32\pi G\left(\frac{p_z^2}{p^2} + \frac{p_z p_{\bar{z}}}{p^2}\right), \quad \langle \overline{f}'(p)\overline{f}'(-p)\rangle_0 = 32\pi G\left(\frac{p_{\bar{z}}^2}{p^2} + \frac{p_z p_{\bar{z}}}{p^2}\right). \tag{142}$$

We now argue that we can drop the $\frac{p_z p_{\bar{z}}}{p^2}$ terms. First, note that in $d = 2$ this term is constant in momentum space, and so corresponds to a delta function contribution to the propagator in position space. Including such delta functions in propagators is equivalent to a redefinition of couplings and operators, since they contract lines down to points, thereby inducing new vertices. The situation in dimensional regularization with $d = 2 + \varepsilon$ is a bit more subtle. While the violation of $p^2 = 4p_z p_{\bar{z}}$ is morally proportional to $\varepsilon$, this can of course be compensated by factors of $1/\varepsilon$ arising from divergent loop integrals. Nonetheless, as shown by explicit computation (see appendix E.4) the effect of including or excluding the $\frac{p_z p_{\bar{z}}}{p^2}$ terms in the

propagator is the same as changing the coupling in front of some local operator. In general, this local operator will be non-Lorentz invariant. We will allow ourselves to add local operators in order to maintain Lorentz invariance, and what we see from the present discussion is that the simplest way to do this is to simply drop the $\frac{p_z p_{\bar{z}}}{p^2}$ terms from the propagators. This should be thought of as part of our renormalization scheme. We therefore take the propagators to be

$$\langle f'(-p) f'(p) \rangle_0 = \quad\underrightarrow{\quad\quad p\quad\quad}\quad = 32\pi G \frac{p_{\bar{z}}^2}{p^2} ,$$

$$\langle \overline{f}'(-p) \overline{f}'(p) \rangle_0 = \quad\text{-----}\underrightarrow{p}\text{-----}\quad = 32\pi G \frac{p_z^2}{p^2} . \tag{143}$$

Arrows indicate momentum flow. With this propagator rule, $f'$ is effectively the same as $\partial_z f$, and $\overline{f}'$ is effectively the same as $\partial_{\bar{z}} \overline{f}$. We then see from (87) that the stress tensor components have indices that match the $\partial_z$ and $\partial_{\bar{z}}$ derivatives that appear. This implies that stress tensor correlators will be Lorentz covariant.

It will also be useful to Fourier transform back to position space. To perform the $d$-dimensional Fourier transform of the propagators (143) is a straightforward application of the integral (240), the result of which produces the position space propagators

$$\langle f'(x) f'(0) \rangle_0 = -8G\pi^{1-\frac{d}{2}} \Gamma\left(\frac{d}{2}+1\right) \frac{\bar{z}^2}{(x \cdot x)^{\frac{d}{2}+1}} ,$$

$$\langle \overline{f}'(x) \overline{f}'(0) \rangle_0 = -8G\pi^{1-\frac{d}{2}} \Gamma\left(\frac{d}{2}+1\right) \frac{z^2}{(x \cdot x)^{\frac{d}{2}+1}} , \tag{144}$$

where $x \cdot x = z\bar{z} + \sum_{i=2}^{d} (x^i)^2$. In $d=2$, (144) becomes

$$\langle f'(x) f'(0) \rangle_0 = -\frac{8G}{z^2} ,$$

$$\langle \overline{f}'(x) \overline{f}'(0) \rangle_0 = -\frac{8G}{\bar{z}^2} . \tag{145}$$

In what follows, we take $\langle f' f' \rangle_0$ and $\langle \overline{f}' \overline{f}' \rangle_0$ as the propagators. When we refer to an "amputated" diagram, we mean that we have divided by these propagators.

## 5.3 Interaction vertex

To the order we work at, there is a single quartic interaction vertex whose Feynman rule reads

$$\begin{array}{c}\text{(vertex diagram)}\end{array} = -\frac{r_c}{64\pi G} . \tag{146}$$

## 5.4 Structure of stress tensor two-point function

The general stress tensor two-point function can be reconstructed from $\langle T_{z\bar{z}}T_{z\bar{z}}\rangle$, as in [33]. To see this, note that Lorentz invariance and parity implies

$$\langle T_{zz}(x)T_{zz}(0)\rangle = \frac{1}{z^4}f_1(y)\,,$$

$$\langle T_{zz}(x)T_{z\bar{z}}(0)\rangle = \frac{1}{z^3\bar{z}}f_2(y)\,,$$

$$\langle T_{zz}(x)T_{\bar{z}\bar{z}}(0)\rangle = \frac{1}{z^2\bar{z}^2}f_3(y)\,,$$

$$\langle T_{z\bar{z}}(x)T_{z\bar{z}}(0)\rangle = \frac{1}{z^2\bar{z}^2}f_4(y)\,,$$

$$\langle T_{\bar{z}\bar{z}}(x)T_{\bar{z}\bar{z}}(0)\rangle = \frac{1}{\bar{z}^4}f_1(y)\,,$$

$$\langle T_{\bar{z}\bar{z}}(x)T_{z\bar{z}}(0)\rangle = \frac{1}{z\bar{z}^3}f_2(y)\,, \tag{147}$$

where the dimensionless variable $y$ is

$$y = \frac{z\bar{z}}{r_c}\,. \tag{148}$$

Stress tensor conservation implies

$$f_1' + y^3\left(\frac{f_2}{y^3}\right)' = 0\,,$$

$$\left(\frac{f_2}{y}\right)' + y\left(\frac{f_3}{y^2}\right)' = 0\,,$$

$$\left(\frac{f_2}{y}\right)' + y\left(\frac{f_4}{y^2}\right)' = 0\,. \tag{149}$$

As $r_c \to 0$, we should recover the usual CFT correlators, which implies that we are looking for solutions with $f_1 \to \frac{c}{2}$ as $y \to \infty$, and with the other functions vanishing in this limit. The central charge $c$ will be computed in terms of $G$ momentarily. Note that $f_3 = f_4$, which implies that $\langle T_{zz}(x)T_{\bar{z}\bar{z}}(y) - T_{z\bar{z}}(x)T_{z\bar{z}}(y)\rangle = 0$. This is compatible with the trace relation $T_{z\bar{z}} = \pi\lambda_{T\bar{T}}\det T$ given that $\langle T_{z\bar{z}}\rangle = 0$.

We find the central charge $c$ by computing correlators at $r_c = 0$, where the stress tensor reads

$$T_{zz}|_{r_c=0} = \frac{1}{8G}\left(f'' - \frac{1}{2}f'^2\right)\,, \quad T_{\bar{z}\bar{z}}|_{r_c=0} = \frac{1}{8G}\left(\bar{f}'' - \frac{1}{2}\bar{f}'^2\right)\,, \quad T_{z\bar{z}}|_{r_c=0} = 0\,. \tag{150}$$

Using (145), we have

$$\langle T_{zz}(x)T_{zz}(0)\rangle|_{r_c=0} = \frac{c/2}{z^4}\,, \quad \langle T_{\bar{z}\bar{z}}(x)T_{\bar{z}\bar{z}}(0)\rangle|_{r_c=0} = \frac{c/2}{\bar{z}^4}\,, \tag{151}$$

with

$$c = \frac{3}{2G} + 1 = c_0 + 1\,. \tag{152}$$

This 1-loop correction to the Brown-Henneaux formula is the same as in [6]. We display the contributing diagrams as

$$\langle T_{zz}(x)T_{zz}(0)\rangle_{r_c=0} = \quad \text{o}\!\!-\!\!\!-\!\!\!-\!\!\text{o} \ + \ \bigcirc \tag{153}$$

where the unfilled circles are used to denote stress tensor insertions.

## 5.5  Correlators of elementary fields

In order to determine any needed counterterms in the action we now consider the 1-loop 4-point and 2-loop 2-point correlators of $(f, \overline{f})$.

### 5.5.1  $\langle f'(p_1)f'(p_2)f'(p_3)f'(p_4)\rangle$

The basic diagram is

$$G_4(p_1, p_2, p_3, p_4) =$$

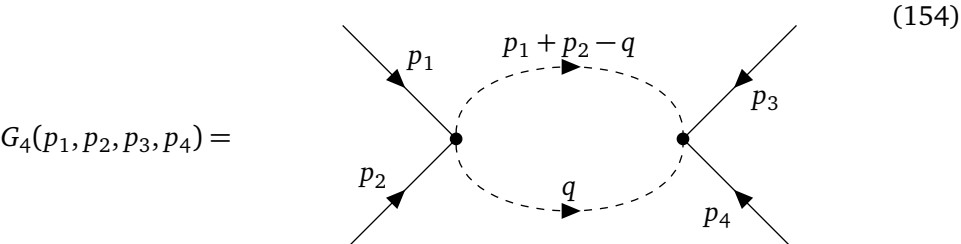

(154)

The full correlator is then

$$\langle f'(p_1)f'(p_2)f'(p_3)f'(p_4)\rangle = G_4(p_1, p_2, p_3, p_4) + G_4(p_1, p_3, p_2, p_4) + G_4(p_1, p_4, p_3, p_2).$$
(155)

The amputated diagram is

$$\begin{aligned}
G_4^{\text{amp}}(p_1, p_2, p_3, p_4) &= 2r_c^2 \int \frac{d^d q}{(2\pi)^d} \frac{(p_{1,\bar{z}} + p_{2,\bar{z}} - q_{\bar{z}})^2 q_{\bar{z}}^2}{(p_1 + p_2 - q)^2 q^2} \\
&= \frac{r_c^2}{12\pi} \frac{(p_{1,\bar{z}} + p_{2,\bar{z}})^4}{(p_1 + p_2)^2}.
\end{aligned}$$
(156)

This diagram is in particular finite, hence requires no $(f')^4$ counterterm.

### 5.5.2  $\langle f'(p_1)f'(p_2)\overline{f}'(p_3)\overline{f}'(p_4)\rangle$

The correlator has an (amputated) tree level contribution

$$\langle f'(p_1)f'(p_2)\overline{f}'(p_3)\overline{f}'(p_4)\rangle_{\text{tree}} = -\frac{r_c}{16\pi G}.$$
(157)

The 1-loop diagram is

$$G_{2,2}(p_1, p_2, p_3, p_4) =$$

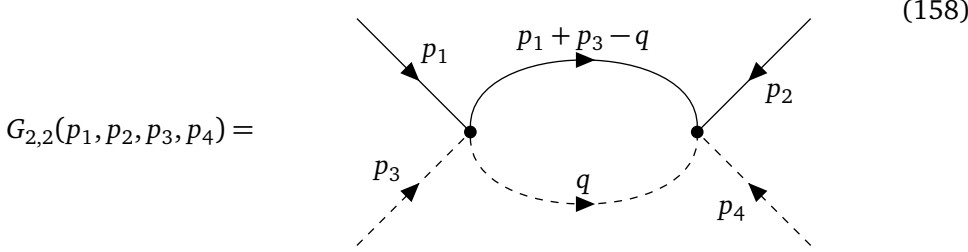

(158)

which we need to evaluate to compute the one-loop contribution to the correlator

$$\langle f'(p_1)f'(p_2)\overline{f}'(p_3)\overline{f}'(p_4)\rangle_{1-\text{loop}} = G_{2,2}(p_1, p_2, p_3, p_4) + G_{2,2}(p_1, p_2, p_4, p_3).$$
(159)

Employing the shorthand $p_{ij} = p_i + p_j$, the result computed using dimensional regularization and setting $d = 2 + \epsilon$ reads

$$G_{2,2}^{\text{amp}}(p_1, p_2, p_3, p_4) = 4r_c^2 \int \frac{d^d q}{(2\pi)^d} \frac{(p_{1,z} + p_{3,z} - q_z)^2 q_{\bar{z}}^2}{(p_1 + p_3 - q)^2 q^2}$$
$$= 4r_c^2 \left[ \frac{p_{13}^2}{32\pi\epsilon} + \frac{6\gamma - 11 - 6\ln(4\pi)}{384\pi} p_{13}^2 + \frac{p_{13}^2}{64\pi} \ln p_{13}^2 \right]. \quad (160)$$

The amputated correlator works out to be

$$\langle f'(p_1) f'(p_2) \overline{f}'(p_3) \overline{f}'(p_4) \rangle_{1-\text{loop}}^{\text{amp}} = \frac{r_c^2 (d+2)(d+4)}{4^{d+5/2} \pi^{\frac{d-3}{2}}} \frac{(p_{13}^2)^{d/2} + (p_{14}^2)^{d/2}}{\sin\left(\frac{\pi d}{2}\right) \Gamma(d/2 + 3/2)}. \quad (161)$$

This has a pole at $\varepsilon = 0$,

$$\langle f'(p_1) f'(p_2) \overline{f}'(p_3) \overline{f}'(p_4) \rangle_{1-\text{loop}}^{\text{amp}} \sim \frac{1}{8\pi\varepsilon} r_c^2 \left[ p_{13}^2 + p_{14}^2 \right]. \quad (162)$$

This divergence is cancelled by the counterterm

$$I_{\text{ct}} = \frac{r_c^2}{4\pi\varepsilon} \int d^2 x \, \partial_z (f' \overline{f}') \partial_{\bar{z}} (f' \overline{f}'). \quad (163)$$

The original action has no term of the form (163). One interpretation is that this implies the existence of a new parameter in our theory corresponding to including an undetermined finite term along with (163). On the other hand, as discussed in the introduction the 3D gravity origin of this theory indicates that no such new parameters should be needed. We thus suspect that the appearance of the undetermined parameter may just reflect the fact that our renormalization scheme has not incorporated all symmetries of the 3D gravity theory.

## 5.6 $\langle f'(x) f'(0) \rangle$ at 2-loops

We will first compute the correlator in momentum space. The relevant Feynman diagram to compute $\langle f'(p) f'(-p) \rangle$ is

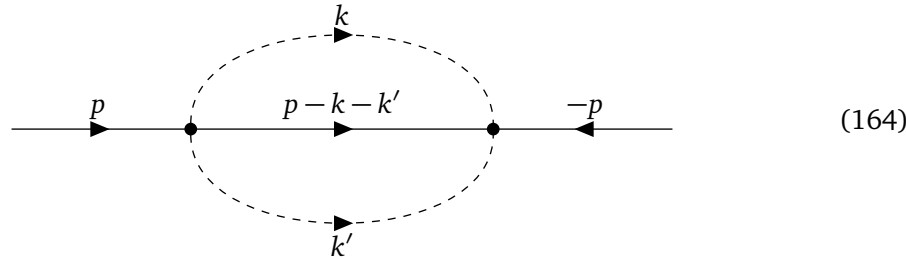

$$\quad (164)$$

The two-loop contribution to the amputated correlator is then

$$\langle f'(p) f'(-p) \rangle_{2-\text{loop}}^{\text{amp}} = 8 \left( \frac{r_c}{64\pi G} \right)^2 (32\pi G)^3 \int \frac{d^2 k}{(2\pi)^2} \int \frac{d^2 k'}{(2\pi)^2} \frac{k_{\bar{z}}^2 k_{\bar{z}}'^2 (p-k-k')_z^2}{k^2 k'^2 (p-k-k')^2}, \quad (165)$$

where the over-all normalization involves two vertex factors as in (146), the normalization of the three internal propagators as in formulas (143), and a symmetry factor of 8. The integrals over the internal momenta $k$ and $k'$ are computed in dimensional regularization in appendix E.3. The result reads

$$\int \frac{d^2 k}{(2\pi)^2} \int \frac{d^2 k'}{(2\pi)^2} \frac{k_{\bar{z}}^2 k_{\bar{z}}'^2 (p-k-k')_z^2}{k^2 k'^2 (p-k-k')^2} = \frac{1}{2^7 3\pi^2} p_z p_{\bar{z}}^3 \log p^2 + \text{polynomial}. \quad (166)$$

Attaching the external legs and Fourier transforming back to position space using formulas (244), we conclude

$$\langle f'(x)f'(0)\rangle_{2-\text{loop}} = -64r_c^2 G^3 \frac{1}{z^4\bar{z}^2} \,. \tag{167}$$

This diagram is in particular finite (up to contact terms at $x = 0$), so no wavefunction renormalization is required.[23]

## 5.7 $\langle T_{z\bar{z}} f' \overline{f}' \rangle$

To identify the need for a counterterm for $T_{z\bar{z}}$, we consider the correlator of the stress tensor with two elementary fields

$$\langle T_{z\bar{z}}(k)f'(p_1)\overline{f}'(p_2)\rangle = \tag{168}$$

The amputated diagram is

$$\langle T_{z\bar{z}}(k)f'(p_1)\overline{f}'(p_2)\rangle_{\text{amp}} = 4\pi r_c^2 \int \frac{d^d q}{(2\pi)^d} \frac{q_z^3 (k_{\bar{z}} - q_{\bar{z}})^3}{q^2 (k-q)^2} = \frac{1}{32\varepsilon} r_c^2 (k^2)^2 + \text{finite} \,. \tag{169}$$

To cancel this divergence we need to redefine this stress tensor component as

$$T_{z\bar{z}} \rightarrow T_{z\bar{z}} - \frac{1}{2\varepsilon} r_c^2 f''' \overline{f}''' \,. \tag{170}$$

Here we have adopted a minimal subtraction scheme. Of course we are free to also add a finite contribution, which will show up below as an undetermined constant in the stress tensor correlator.

## 5.8 $\langle T_{z\bar{z}} T_{z\bar{z}} \rangle$

To compute $\langle T_{z\bar{z}} T_{z\bar{z}} \rangle$ to 2-loop order, we recall

$$4G T_{z\bar{z}} = -\frac{1}{4} r_c f'' \overline{f}'' + \frac{1}{8} r_c (f'' \overline{f}'^2 + f'^2 \overline{f}'') - \frac{1}{8} r_c^2 (f''' \overline{f}'' \overline{f}' + f' f'' \overline{f}''') + \text{quartic} \,. \tag{171}$$

The contributing diagrams to 2-loop order are

$$\langle T_{z\bar{z}}(x) T_{z\bar{z}}(0)\rangle = \tag{172}$$

The first three diagrams are trivially computed by Wick contraction in position space. The 1-loop diagram is

$$= \frac{9r_c^2}{z^4\bar{z}^4} \,, \tag{173}$$

---

[23]As can be seen in (E.3), the integral (166) does have a divergence in dimensional regularization. However, the divergence is a polynomial in the momentum, which only leads to delta function contact terms in position space.

and the two simple 2-loop diagrams sum to

$$\text{(diagrams)} = \frac{12Gr_c^2}{z^4\bar{z}^4} - \frac{192Gr_c^3}{z^5\bar{z}^5} + \frac{1200Gr_c^4}{z^6\bar{z}^6} \, . \tag{174}$$

We next turn to the 2-loop diagram in (172). Working in momentum space, the contribution to $\langle T_{z\bar{z}}(-k)T_{z\bar{z}}(k)\rangle$ is

$$\langle T_{z\bar{z}}(-k)T_{z\bar{z}}(k)\rangle_\infty = \text{(diagram)} = 2^8\pi^3 Gr_c^3 \left[\int \frac{d^d p}{(2\pi)^d} \frac{p_z^3(k_{\bar{z}}+p_{\bar{z}})^3}{p^2(k+p)^2}\right]^2 \, . \tag{175}$$

This diagram has double and single pole divergences in $\varepsilon$. The double pole is polynomial in $k$, hence can be ignored as it won't contribute to the 2-point function at finite spatial separation. The simple pole is cancelled, by design, via the stress tensor counterterm (170); i.e. by the two 1-loop diagrams in which one of the stress tensor insertions is given by the counterterm in (170). The resulting finite part is

$$\langle T_{z\bar{z}}(-k)T_{z\bar{z}}(k)\rangle_\infty = 2^{-7}\pi r_c^3 G\left(a\ln k^2 + (\ln k^2)^2\right)(k^2)^4 + \text{polynomial} \, . \tag{176}$$

The constant $a$ has been left unspecified since it can be shifted arbitrarily due to the freedom in including a finite counterterm in (170). Fourier transforming back to position space, we obtain

$$\langle T_{z\bar{z}}(x)T_{z\bar{z}}(0)\rangle_\infty = 2^8 \cdot 3^2 r_c^3 G \frac{\ln(\mu^2 z\bar{z})}{(z\bar{z})^5} \, , \tag{177}$$

where we now traded the arbitrary constant $a$ for a renormalization scale $\mu$. [24]

## 5.9   Summary

Combining results, to 2-loop order we have found

$$\langle T_{z\bar{z}}(x)T_{z\bar{z}}(0)\rangle = \frac{3}{(z\bar{z})^2}\left[(3+4G)\left(\frac{r_c}{z\bar{z}}\right)^2 - 64G\left(1-12\ln(\mu^2 z\bar{z})\right)\left(\frac{r_c}{z\bar{z}}\right)^3 + 400G\left(\frac{r_c}{z\bar{z}}\right)^4\right] \, . \tag{178}$$

Using the Ward identities (147) and (149), we read off the other 2-point functions

$$\langle T_{zz}(x)T_{zz}(0)\rangle = \frac{1}{z^4}\left[\frac{c}{2} + 10(3+4G)\left(\frac{r_c}{z\bar{z}}\right)^2 + 96G\left(8+60\ln(\mu^2 z\bar{z})\right)\left(\frac{r_c}{z\bar{z}}\right)^3 + 2520G\left(\frac{r_c}{z\bar{z}}\right)^4\right] \, ,$$

$$\langle T_{zz}(x)T_{z\bar{z}}(0)\rangle = \frac{4}{z^3\bar{z}}\left[-(3+4G)\left(\frac{r_c}{z\bar{z}}\right)^2 + 24G\left(1-30\ln(\mu^2 z\bar{z})\right)\left(\frac{r_c}{z\bar{z}}\right)^3 - 360G\left(\frac{r_c}{z\bar{z}}\right)^4\right] \, ,$$

$$\langle T_{zz}(x)T_{\bar{z}\bar{z}}(0)\rangle = \frac{3}{(z\bar{z})^2}\left[(3+4G)\left(\frac{r_c}{z\bar{z}}\right)^2 - 64G\left(1-12\ln(\mu^2 z\bar{z})\right)\left(\frac{r_c}{z\bar{z}}\right)^3 + 400G\left(\frac{r_c}{z\bar{z}}\right)^4\right] \, , \tag{179}$$

where $c = c_0 + 1 = \frac{3}{2G} + 1$.

---

[24]Logarithms also appear in the $T\bar{T}$ deformed correlation functions of [33,34].

## 6 Discussion

The main results of this paper are twofold. We first of all gave evidence for the Nambu-Goto action (in Hamiltonian form) as the all orders action for 3D gravity with a cutoff planar boundary. Second, we used the action to compute correlators of the stress tensor operator to two-loop order. Our proposal for the action was based on finding a suitable field redefinition yielding Nambu-Goto up eighth order in fields. It would of course be desirable to prove this conjecture and determine the explicit form of the field redefinition to all orders. Although the action takes the familiar Nambu-Goto form, the stress tensor is not the canonical one, which is due to the way that the original translation symmetries of the AdS$_3$ background act on the redefined fields. Our computation of stress tensor correlators to two-loop order revealed the need for one stress tensor counterterm, with an associated undetermined finite part. As discussed in the introduction, given the general arguments for the renormalizability of pure 3D gravity, including the case of a finite planar cutoff boundary, we expect that all parameters should be fixed by symmetries. The implementation of these symmetries is complicated by the non-Lorentz invariant form of the action and by the nonlocal field redefinition that puts the action in Nambu-Goto form. A task for the future is to systematically implement the Ward identities corresponding to these symmetries and check if these yield unique results for stress tensor correlators. The ultimate goal here is to get sufficient control over the stress tensor correlators to say something about their short distance structure, since this gets to the heart of the nature of this theory, including its anticipated nonlocal character; e.g [64, 65].

It would also be worthwhile to further develop cases with curved cutoff boundaries. We considered the Chern-Simons computation of the action for a finite $S^2$ boundary, and it should be possible to extend this to 1-loop and compare with results in [66]; see also [67, 68] for related results. The technical complication here is the two patches needed to properly define the gauge connections on the sphere.

We close by commenting on the appearance of the Nambu-Goto action in our analysis. By construction, solutions of our Nambu-Goto equations of motion yield flat two-dimensional surfaces embedded in AdS$_3$. On the other hand, the precise Nambu-Goto action that arises is that of a string worldsheet embedded in flat $\mathbb{R}^3$, with $\alpha'$ controlled by $r_c$. We usually think of the solutions as describing extremal area surfaces embedded in this flat spacetime. Apparently, there is a correspondence between flat surfaces embedded in AdS$_3$ and extremal area surfaces embedded in $\mathbb{R}^3$.

## Acknowledgements

We thank Jan de Boer, Alejandra Castro, Konstantinos Roumpedakis and Michael Ruf for useful discussions. P.K. and R.M. are supported in part by the National Science Foundation under grant PHY-1914412. E.H. acknowledges support from the Gravity Initiative at Princeton University.

## A   CS action with cutoff sphere boundary, and appearance of Weyl anomaly

The maximally symmetric solution to the 3D Einstein equations with a negative cosmological constant in Euclidean signature has the topology of a solid sphere. Its metric can be written as

$$ds^2 = d\eta^2 + \sinh^2 \eta \, d\Omega_2^2 \,, \tag{180}$$

where $\eta \geq 0$ and $d\Omega_2^2 = d\theta^2 + \sin^2\theta\, d\varphi^2$ is the metric of the 2-sphere conformal boundary. In this section, we calculate the classical action of this geometry using both the metric and Chern–Simons language, and show how the Weyl anomaly emerges. Our analysis differs from the one in [6], where the Weyl anomaly appeared as a logarithmic divergence of the boundary action near the poles. Instead, we work with two coordinate patches and see the Weyl anomaly appear from the nontrivial relation between the gauge connections on each patch.

## A.1 Metric calculation

The on-shell value of the Einstein–Hilbert action (6) can be calculated using $R = 6\Lambda = -6$

$$I_{\text{EH}} = -\frac{1}{8G}\int_0^{\eta_c} d\eta\,(-4\sinh^2\eta) = -\frac{1}{4G}(2\eta_c - \sinh 2\eta_c)\,. \tag{181}$$

To calculate the boundary action in (9), we first relate $\eta$ to the Fefferman–Graham coordinate $r$ defined in (8) by $2\eta = -\ln r$. Observing that $K = \frac{1}{2}g^{ij}\partial_\eta g_{ij}$, and $\sqrt{\det g_{ij}}R(g_{ij}) = 2\sin\theta$, we obtain

$$I_{\text{bndy}} = -\frac{1}{2G}\Big[\sinh^2\eta_c(2\coth\eta_c - 1)\Big] + \frac{\eta_c}{2G}\,. \tag{182}$$

As expected, both the exponential and the linear divergences in $\eta_c$ cancel with $I_{\text{EH}}$

$$I = I_{\text{EH}} + E_{\text{bndy}} = -\frac{1}{4G}(1 - e^{-2\eta_c})\,. \tag{183}$$

The term linear in $\eta_c$ is logarithmic in $r_c$ and cannot be cancelled by adding to the action a covariant local boundary term. Instead, we used the second term in (9), which is proportional to $\eta_c$ times the Ricci curvature of the boundary. Such a term is not covariant since it depends explicitly on the coordinate value $\eta_c$. Indeed, this term signals the presence of a Weyl anomaly in the CFT, and manifests itself on the gravity side as the absence of diffeomorphism invariance.

## A.2 Chern–Simons calculation

In the previous section, it was not necessary to chose explicit coordinates on the boundary two-sphere to do this calculation. Indeed the action only depended on its overall area. The fact that $S^2$ cannot be covered in a single coordinate patch did not pose any problems. We will need to face this issue in now to do the analogous Chern–Simons calculation.

### A.2.1 Stereographic projection

It is possible to cover all of the sphere except for one point using the stereographic projection. We will define the complex coordinate

$$z_{\text{S}} = \cot(\theta/2)e^{i\varphi}\,, \tag{184}$$

which is regular everywhere but the north pole at $\theta = 0$ and in terms of which two-sphere metric is

$$d\Omega_2^2 = \frac{4dz_{\text{S}}\,d\bar{z}_{\text{S}}}{(1 + z_{\text{S}}\bar{z}_{\text{S}})^2}\,. \tag{185}$$

Similarly, we can cover all but the south pole using

$$z_{\text{N}} = z_{\text{S}}^{-1} = \tan(\theta/2)e^{-i\varphi}\,, \tag{186}$$

which gives the same metric as before and is related to $z_{\text{S}}$ by a rotation of the sphere that maps the north to the south pole: $(\theta, \varphi) \to (\pi - \theta, -\varphi)$. We can choose a local Lorentz frame for

which the associated zweibein and spin connection, which has only a single component in two dimensions, are[25]

$$e_{S,N}^+ = \frac{-i\,dz}{1+z\bar{z}}\,, \qquad\qquad e_{S,N}^- = \frac{-i\,d\bar{z}}{1+z\bar{z}}\,, \qquad\qquad \omega_{S,N} = -\frac{z\,d\bar{z} - \bar{z}\,dz}{1+z\bar{z}}\,, \qquad (188)$$

where $z$ is either $z_S$ or $z_N$. In terms of the original variables, this gives

$$e_S^+ = \frac{1}{2}e^{i\varphi}(id\theta + \sin\theta\,d\varphi)\,, \qquad\qquad e_N^+ = -\frac{1}{2}e^{-i\varphi}(id\theta + \sin\theta\,d\varphi)\,,$$
$$e_S^- = \frac{1}{2}e^{-i\varphi}(id\theta - \sin\theta\,d\varphi)\,, \qquad\qquad e_N^- = -\frac{1}{2}e^{i\varphi}(id\theta - \sin\theta\,d\varphi)\,,$$
$$\omega_S = 2i\cos^2(\tfrac{\theta}{2})d\varphi\,, \qquad\qquad \omega_N = -2i\sin^2(\tfrac{\theta}{2})d\varphi\,. \qquad (189)$$

Using these coordinates, the 3-dimensional Chern–Simons gauge connections on each patch are

$$A = (\omega + d\eta)L_0 + e^{\eta}e^+ L_1 - e^{-\eta}e^- L_{-1}\,,$$
$$\bar{A} = (\omega - d\eta)L_0 + e^{-\eta}e^+ L_1 - e^{\eta}e^- L_{-1}\,. \qquad (190)$$

### A.2.2 Action

We are now ready to calculate the on-shell action in the Chern–Simons language. It consists of two terms, the bulk Einstein–Hilbert action (6) and the boundary contribution (9).

Starting with the Einstein–Hilbert action, we can rewrite it in terms of Chern–Simons gauge connections as follows[26]

$$I_{\text{EH}} = -\frac{1}{16\pi G}\int_M d^3x\sqrt{g}(R - 2\Lambda) \qquad (191)$$
$$= -\frac{ik}{4\pi}\int_M \text{Tr}(A\wedge dA + \tfrac{2}{3}A^3) + \frac{ik}{4\pi}\int_M \text{Tr}(\bar{A}\wedge d\bar{A} + \tfrac{2}{3}\bar{A}^3) + \frac{ik}{4\pi}\int_M d(\text{Tr}\,A\wedge\bar{A})\,,$$

where $k = 1/4G$. We will split up the integral over the manifold $M$ into a contribution from $\text{AdS}_N$ and $\text{AdS}_S$ as depicted in Figure 1. To evaluate the first term, we can use the explicit form of the gauge potentials (190)

$$I_{\text{CS}}[A] = -\frac{ik}{8\pi}\int_0^{\eta_c} d\eta\left(\int_{\text{AdS}_S} d\omega_S + \int_{\text{AdS}_N} d\omega_N\right) = -\frac{k}{2}\eta_c\,. \qquad (192)$$

One can check that the result does not depend on the location of the disk $D$ which separates the two patches, as long as it does not cross either of the poles. The second term in (191) yields the same contribution, $-I_{\text{CS}}[\bar{A}] = -k\eta_c/2$. The total derivative in the third term of (191) will contribute not only on the cutoff boundary $S^2 = S_S^2 \cup S_N^2$ at $\eta_c$ but also on the internal boundary $D$ that separates northern from the southern hemisphere,

$$\frac{ik}{4\pi}\int_M d(\text{Tr}\,A\wedge\bar{A}) = \frac{ik}{4\pi}\int_{S^2}\text{Tr}\,A\wedge\bar{A} - \frac{ik}{4\pi}\int_D \text{Tr}(A_N\wedge\bar{A}_N - A_S\wedge\bar{A}_S)\,. \qquad (193)$$

---

[25]In Euclidean signature, the flatness condition (103) contains additional minus signs,

$$de^+ + \omega\wedge e^+ = de^- - \omega\wedge e^- = 0\,. \qquad (187)$$

This can be traced back to the minus sign in the Lorentzian identity $\epsilon_{\mu\nu\rho}\epsilon^{\mu\sigma\tau} = -\delta_{\nu\rho}^{\sigma\tau}$, whereas that minus sign is absent in Euclidean signature.

[26]There is an additional factor of $i$ in the relation between these actions because we now work in Euclidean signature. We will not change the gauge group with respect to the main text, but rather include explicit factors of $i$ in the gauge connections.

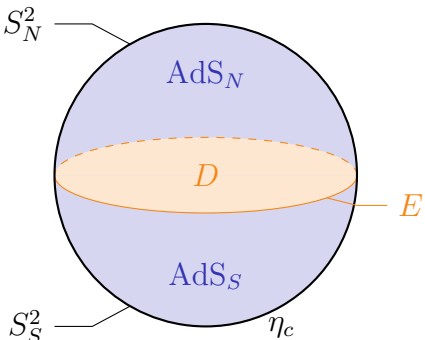

Figure 1: Global Euclidean $AdS_3$ and its different components and boundaries. The bulk of $AdS_3$ is composed of two patches, $AdS_S$ and $AdS_N$, whose boundaries at $\eta_c$ are the southern and a northern hemisphere $S^2_S$ and $S^2_N$, respectively. The bulk components are separated by an equatorial disk $D$, whereas the boundary hemispheres touch at the equator $E$.

The signs are fixed by comparing the volume form in the bulk, which we took $\propto d\eta \wedge d\theta \wedge d\varphi$, with the one on $S^2$ that we choose $\propto d\theta \wedge d\varphi$ and on the disk $D$ which we fix to be $\propto d\eta \wedge d\varphi$. There is an additional sign for $A_S \wedge \overline{A}_S$ coming from the outward pointing normal $n_i dx^i = -d\theta$. Calculate from (190) that $\text{Tr}A \wedge \overline{A} = d\eta \wedge \omega + 2\sinh(2\eta)e^+ \wedge e^-$ and pulling this back to each of the boundaries, we find

$$\frac{ik}{4\pi}\int_M d(\text{Tr}A \wedge \overline{A}) = \frac{ik}{2\pi}\sinh(2\eta_c)\int_{S^2}\frac{-i}{2}\sin\theta \, d\theta \wedge d\phi + \frac{ik}{4\pi}\int_D 2i \, d\eta \wedge d\varphi$$
$$= k\sinh(2\eta_c) - k\,\eta_c \,. \tag{194}$$

Altogether, we find

$$I_{\text{EH}} = -k(2\eta_c - \sinh(2\eta_c)) \,, \tag{195}$$

which agrees with the metric calculation (181).

What remains to be calculated is the boundary action $I_{\text{bndy}}$ (9) on the $S^2$ boundary of $M$. We explain in appendix D how to express the extrinsic curvature in terms of the Chern–Simons gauge connections. In Euclidean signature, the result reads

$$I_{\text{bndy}} = -\frac{k}{2\pi}\int_{S^2} d^2x \sqrt{\det g_{ij}}(\partial_a n^a - 1) - \frac{ik}{2\pi}\int_{S^2}\text{Tr}(A \wedge \overline{A}) \,. \tag{196}$$

In the case of interest, $\partial_a n^a$ vanishes and we choose a gauge for which the $L_0$ component of the gauge connections is normal to the boundary and the other components are parallel. The action the simplifies to

$$I_{\text{bndy}} = -\frac{ik}{4\pi}\int_{S^2}\text{Tr}\left[2A \wedge \overline{A} - L_0(A - \overline{A}) \wedge (A - \overline{A})\right] \,. \tag{197}$$

The first term was already calculated in (193) (indeed it adds up with (197) to give the boundary contribution given in (93)). The second one only depends on the boundary frame field,

$$I_{\text{bndy}} = -2k[\sinh(2\eta_c) - \sinh^2(\eta_c)] \,, \tag{198}$$

which agrees with (182).

# B    Chern-Simons action as a boundary term

In order to reduce the action to a boundary term, we start by implementing the space-time split (111) and making use of the constraints $\tilde{\mathcal{F}} = \tilde{\bar{\mathcal{F}}} = 0$. Here

$$
\begin{aligned}
S[A,\overline{A}] = & \frac{k}{4\pi} \int_{\mathcal{M}} \mathrm{Tr}\Big[\tilde{A} \wedge dt \wedge \partial_t \tilde{A} - \tilde{\overline{A}} \wedge dt \wedge \partial_t \tilde{\overline{A}}\Big] \\
& + \frac{k}{4\pi} \int_{\mathcal{M}} \mathrm{Tr}\Big[-\tilde{d}\big(\tilde{A} \wedge A_t \, dt\big) + \tilde{d}\big(\tilde{\overline{A}} \wedge \overline{A}_t \, dt\big)\Big] + I_{\mathrm{bndy}} \,.
\end{aligned}
\tag{199}
$$

The second line of (199) involves boundary terms that can easily be evaluated in terms of $g$ and $\overline{g}$. The first line is a bulk term, which when evaluated on the flat connections (114) reads

$$
\frac{k}{4\pi} \int_{\mathcal{M}} \mathrm{Tr}\big[\tilde{A} \wedge dt \wedge \partial_t \tilde{A}\big] = \frac{k}{4\pi} \int_{\mathcal{M}} \mathrm{Tr}\Big[-\frac{1}{3}\big(g^{-1}dg\big)^3 - \tilde{d}\big(g^{-1}\partial_t g \wedge g^{-1}\tilde{d}g\big)\Big],
\tag{200}
$$

and similarly for the barred connections. The second term on the right hand side of (200) is already a boundary term. The first term is a Wess-Zumino term, which becomes a boundary term once an explicit parametrization for the group element $g$ is chosen. For the Gauss parametrization (115), one finds

$$
\frac{k}{4\pi} \int_{\mathcal{M}} \mathrm{Tr}\Big[-\frac{1}{3}\big(g^{-1}dg\big)^3\Big] = -\frac{k}{4\pi} \int_{\partial\mathcal{M}} \lambda^2 \wedge d\Psi \wedge dF \,.
\tag{201}
$$

Combining equations (201), (200), and (199) yields an expression for the full action written as a boundary term. Its expression as a functional of the Gauss parameters $(\lambda, \overline{\lambda}, \Psi, \overline{\Psi}, F, \overline{F})$ has been written in the main text in equation (118).

# C    Relation between Chern-Simons theory at finite cut-off and coupling to topological gravity

The objective of this appendix is to connect the ideas of AdS$_3$ gravity with a finite cut-off and the $T\overline{T}$ deformation of a conformal field theory as described by coupling to topological gravity; see [69–72] for relevant background. In this appendix, we follow the conventions in [72], with $g_{ij} = \delta_{ab} e_i^a e_j^b$.

The topological gravity formulation is based on the observation that the $T\overline{T}$ flow equation for the deformed action

$$
\frac{dI_{\lambda_{T\overline{T}}}}{d\lambda_{T\overline{T}}} = -\frac{1}{4} \int d^2x \, \sqrt{g}\det T_j^i \,,
\tag{202}
$$

can be solved by defining an action with auxiliary fields that will be integrated out. In particular, we define

$$
I_{\lambda_{T\overline{T}}} = I_{\mathrm{grav}}[\tilde{e}^a{}_i, e_i^a] + I_0[e_i^a, \psi] \,,
\tag{203}
$$

where $\psi$ and $\tilde{e}_i^a$ are the original matter fields and veilbein in the undeformed theory, and $e_i^a$ is the vielbein of the deformed theory. The action $I_0$ is the undeformed action, while the topological gravity action reads

$$
I_{\mathrm{grav}} = \frac{1}{4\pi^2\lambda_{T\overline{T}}} \int d^2x \, \epsilon^{ij} \epsilon_{ab} (e - \tilde{e})_i^a (e - \tilde{e})_j^b \,.
\tag{204}
$$

In this appendix we take $|\epsilon^{ij}| = |\epsilon_{ab}| = 1$ and also write $e = \det e_i^a$. In order to obtain the action for the fields $\psi$ coupled to the background vielbein $e$, the prescription is to path integrate over $\tilde{e}_i^a$, which in the classical limit can be performed by extremizing (203) with respect to $\tilde{e}$.

An important ingredient is the stress tensor of the deformed theory,

$$T_a^i = \frac{2\pi}{e} \frac{\delta I_{\lambda_{T\overline{T}}}}{\delta e_i^a} = -\frac{1}{\pi \lambda_{T\overline{T}}} \epsilon^{ij} \epsilon_{ab} (\tilde{e}_j^b - e_j^b). \tag{205}$$

As we will see momentarily, this formula will be recovered from the boundary conditions imposed on Chern-Simons connections at a finite radial cut-off.

In section 4.4, we reduced the Chern-Simons action to a boundary action depending on Gauss parameters $(\lambda, \overline{\lambda}, \Psi, \overline{\Psi}, F, \overline{F})$. Boundary conditions (117) and (116) can be thought of as providing solutions for $(\lambda, \overline{\lambda}, \Psi, \overline{\Psi})$ in terms of $F$ and $\overline{F}$. The resulting action is then a functional of $F$ and $\overline{F}$. However, in practice, this procedure cannot be carried out analytically. Having noted this limitation, the boundary conditions equations have a beautiful interpretation: they coincide with (205).

In order to see this, we compute the boundary stress tensor

$$T_a^i \equiv \frac{2\pi}{e} \frac{\delta I}{\delta e_i^a}, \tag{206}$$

in terms of the Gauss parameters. The time components read

$$T_+^t = -\frac{k}{r_c e} \left( \Psi' + \lambda^2 \Psi^2 F' + 2\Psi \frac{\lambda'}{\lambda} \right), \quad T_-^t = \frac{k}{r_c e} \left( \overline{\Psi}' + \overline{\lambda}^2 \overline{\Psi}^2 \overline{F}' + 2\overline{\Psi} \frac{\overline{\lambda}'}{\overline{\lambda}} \right). \tag{207}$$

Using these formulas, the differential equations imposed by the boundary conditions (117) can be written as

$$\begin{aligned}
e_x^+ - \lambda^2 F' + \frac{r_c}{k} \epsilon_{xi} \epsilon^{+a} T_a^i &= 0, \\
e_x^- - \overline{\lambda}^2 \overline{F}' + \frac{r_c}{k} \epsilon_{xi} \epsilon^{-a} T_a^i &= 0.
\end{aligned} \tag{208}$$

These are precisely the spatial components of equation (205) upon making the identification

$$\lambda^2 = \frac{\tilde{e}_x^+}{F'}, \quad \overline{\lambda}^2 = \frac{\tilde{e}_x^-}{\overline{F}'}, \tag{209}$$

and

$$\frac{r_c}{k} = \pi \lambda_{T\overline{T}}, \quad \text{or} \quad r_c = \frac{\pi}{6} \lambda_{T\overline{T}} c. \tag{210}$$

Formulas (208) only capture the definition of the time components of the deformed stress tensor. The space components in terms of the connections (96) read explicitly

$$\begin{aligned}
T_-^x &= -\frac{1}{\tilde{e}} \frac{k}{r_c} \overline{A}_t^+|_{r_c} = -\frac{1}{\tilde{e}} \frac{k}{r_c} f_t^+, \\
T_+^x &= \frac{1}{\tilde{e}} \frac{k}{r_c} A_t^-|_{r_c} = \frac{1}{\tilde{e}} \frac{k}{r_c} f_t^-.
\end{aligned} \tag{211}$$

The time components of the connections must obey the boundary condition (98), which we repeat here

$$(E - f)_t^\pm|_{r_c} = \pm e_t^\pm. \tag{212}$$

We can think of this boundary condition as fixing $E^\pm$ in terms of the fixed boundary vielbein $e$ and the one-forms $f^\pm$, which remain unfixed. Even though the time components of $f^\pm$ remain

unfixed, they do not appear explicitly in the reduced action, given that the time components of the connection are Lagrange multipliers. However, a physical meaning can be attributed to $f_t^{\pm}$. For this, we introduce the time components of an undeformed vielbein $\tilde{e}_t^{\pm}$, and relabel as follows

$$f_t^{\pm} = \tilde{e}_t^{\pm} - e_t^{\pm}. \tag{213}$$

The holographic stress tensor formula can then be recast in terms of $\tilde{e}_t^{\pm}$ instead of $f_t^{\pm}$. The result is

$$
\begin{aligned}
e_t^+ - \tilde{e}_t^+ + \frac{r_c}{k}\epsilon_{ti}\epsilon^{+a}\tilde{T}_a^i &= 0, \\
e_t^- - \tilde{e}_t^- + \frac{r_c}{k}\epsilon_{ti}\epsilon^{-a}\tilde{T}_a^i &= 0.
\end{aligned}
\tag{214}
$$

These are precisely the time components of the definition of the deformed stress tensor in a $T\overline{T}$ deformed theory. In summary, we conclude that the Dirichlet boundary conditions imposed at $r = r_c$ together with the definition of the holographic stress tensor have a nice interpretation in the context of a theory deformed by a coupling to topological gravity. This is achieved by identifying our Gauss parameters $\lambda$ and $\overline{\lambda}$ with the space components of an undeformed zweibein $\tilde{e}^{\pm}$ as written in (209), as well as identifying the time components of $f^{\pm}$ with the time components of such zweibein as written in (213). The fixed zweibein $e$ at $r = r_c$ plays the role of the deformed zweibein.

The connection between Chern-Simons theory with finite cutoff and $T\overline{T}$ deformed CFT is even more apparent at the level of the action. Evaluating the action in terms of $F$, $\overline{F}$, and the zweibein $\tilde{e}^{\pm}$ introduced here as a relabeling of $\lambda$, $\overline{\lambda}$, and $f_t^{\pm}$, we find

$$I[F, \overline{F}, \tilde{e}^{\pm}; e] = I_0[F, \overline{F}; \tilde{e}^{\pm}] + I_{\text{grav}} + I_{\text{extra}}. \tag{215}$$

The first term is simply the Wick rotated action (118) we found in the main text when studying the reduced action of AdS$_3$ gravity with a curved background at $r = 0$. The second term is a coupling between $e$ and $\tilde{e}$. Explicitly,

$$I_{\text{grav}} = \frac{1}{16\pi G r_c}\int d^2x\, \epsilon^{ij}\epsilon_{ab}(e - \tilde{e})_i^a(e - \tilde{e})_j^b. \tag{216}$$

This matches the topological coupling (203) introduced above as a mechanism to deform the original theory by the $T\overline{T}$ operator. The last term in (215) reads

$$I_{\text{extra}} = \frac{1}{16\pi G}\int d^2x \det(e)\frac{(\tilde{\omega}_x(e) - \omega_x(\tilde{e}))^2}{\tilde{e}_x^+\tilde{e}_x^-}. \tag{217}$$

We now show that this term vanishes on-shell and so does not affect the value of the deformed stress tensor. We do so by computing the flatness equations of the Chern-Simons theory at the cutoff boundary $r = r_c$. We relabel the parameters $\lambda$, $\overline{\lambda}$ by introducing the space components of an undeformed zweibein $\tilde{e}_x$, as explained in formulas (209). We also relabel $f_t^{\pm}$ in terms of $\tilde{e}_t$ as written in (213). We therefore expect the on-shell conditions at $r = r_c$ to involve the zweibeins $\tilde{e}$ and $e$, the functions $F$ and $\overline{F}$, and the spin connection at the Dirichlet boundary $\omega$.

Interestingly, when using the boundary conditions (117), the field strength components do not depend on $F$ and $\overline{F}$ explicitly. They involve exclusively $\tilde{e}$, $e$, and $\omega$. Explicitly, the

components of the field strength evaluate to the following:

$$\begin{aligned}
\text{Tr}\left[(\mathcal{F}-\overline{\mathcal{F}})_{xt}L_{\mp 1}\right] &= \left(e^{\pm}\wedge\omega\pm de^{\pm}\right)_{tx}, \\
-\text{Tr}\left[\overline{\mathcal{F}}_{xt}L_1\right] &= \left(\tilde{e}^{-}\wedge\omega - d\tilde{e}^{-}\right)_{tx}, \\
\text{Tr}\left[\mathcal{F}_{xt}L_{-1}\right] &= \left(\tilde{e}^{+}\wedge\omega + d\tilde{e}^{+}\right)_{tx}, \\
2r_c\,\text{Tr}\left[\mathcal{F}_{xt}L_0\right] &= \left(2\tilde{e}^{+}\wedge(e^{-}-\tilde{e}^{-}) - d\omega\right)_{tx}, \\
2r_c\,\text{Tr}\left[\overline{\mathcal{F}}_{xt}L_0\right] &= \left(2\tilde{e}^{-}\wedge(e^{+}-\tilde{e}^{+}) - d\omega\right)_{tx}.
\end{aligned} \tag{218}$$

An important feature of the first three lines in formulas (218) is that on-shell, the zweibein $\tilde{e}$ obeys the relation

$$\omega_x(e) \equiv \frac{1}{e}\tilde{\epsilon}^{ij}\partial_i e_{j,a}e_x^a = \frac{1}{\tilde{e}}\tilde{\epsilon}^{ij}\partial_i\tilde{e}_{j,a}\tilde{e}_x^a \equiv \omega_x(\tilde{e}). \tag{219}$$

This implies (217) vanishing.

To summarize, in this appendix, we showed that Chern-Simons theory with curved cutoff boundary can be understood on-shell as coupling the theory at an asymptotic boundary at $r = 0$ to topological gravity. While conceptually satisfying, the action (203) is not very practical for direct computation because the boundary conditions (117) cannot be solved analytically.

# D Gibbons-Hawking-York Term in Chern-Simons

Here we will show how the Gibbons-Hawking-York (GHY) term can be written in terms of the Chern-Simons variables $A$, and $\overline{A}$. As an intermediate step, we will first write this term in terms of the vielbein and spin connection. Though the Chern-Simons description only applies in 3 dimensions, translating from the metric to vielbein description is not simplified in 3 dimensions, so we perform that portion of the calculation in arbitrary dimension.[27]

On a $D = d + 1$ dimensional spacetime $M$, the GHY term is given by

$$S_{\text{bndy}} = -\frac{1}{16\pi G}\int_{\partial M} 2\sqrt{h}d^d x K, \tag{220}$$

where $h$ is the induced metric on the boundary and $K$ is the trace of the extrinsic curvature. If $n$ is the outward-pointing normal to $\partial M$ normalized so $n^\mu n_\mu \equiv \sigma = \pm 1$, then $K_{\mu\nu} = h_\mu^\lambda\nabla_\lambda n_\nu$ so by writing the projection down to $\partial M$ as $h_\nu^\mu = \delta_\nu^\mu - \sigma n^\mu n_\nu$, we obtain the identity

$$K = g^{\mu\nu}K_{\mu\nu} = \nabla^\mu n_\mu - \sigma n^\mu n^\nu\nabla_\mu n_\nu. \tag{221}$$

The final term here is equal to $\frac{1}{2}\sigma n^\nu\nabla_\nu(n^\mu n_\mu)$, which is zero so long as we choose an extension of $n_\mu$ off $\partial M$ which is everywhere normalized. We will assume here that we have chosen such an extension.

Using lower-case Latin letters for flat Lorentz indices we may write $\nabla_\mu n^\mu = \nabla_a n^a = e_a^\mu\nabla_\mu n^a$ so

$$K = e_a^\mu\left(\delta_b^a\partial_\mu + \omega_{\mu b}^a\right)n^b = \partial_a n^a + e_a^\mu\omega_{\mu b}^a n^b. \tag{222}$$

---

[27]Throughout this appendix we work in Euclidean signature, but to obtain the Lorentzian result it is sufficient to negate the overall sign of (220) which propagates to negating the overall sign of the final results, (223), (227), and (230).

The second term admits a nice coordinate-independent representation in terms of the vielbein and spin connection, leading us to

$$S_{\text{bndy}} = -\frac{1}{16\pi G} \int_{\partial M} 2\sqrt{h} d^d x \left( e_a^\mu \omega_{\mu b}^a n^b + \partial_a n^a \right)$$

$$= -\frac{1}{16\pi G} \int_{\partial M} \left[ -\frac{1}{(d-1)!} \epsilon_{abc_2...c_d} \omega^{ab} \wedge e^{c_2} \wedge \ldots \wedge e^{c_d} + 2\sqrt{h} d^d x \partial_a n^a \right]. \quad (223)$$

To show this final equality, it is sufficient to note that $\omega^{ab} = e_c^\mu \omega_\mu^{ab} e^c$ and the identity

$$\int_{\partial M} e^c \wedge e^{c_2} \wedge \ldots \wedge e^{c_d} = \int_{\partial M} n_d \epsilon^{dcc_2...c_d} \sqrt{h} d^d x, \quad (224)$$

which follows by antisymmetry of the wedge and the observation that $e^c \wedge e^{c_2} \wedge \ldots \wedge e^{c_d}$ pulled back to the boundary should annihilate the normal to the boundary.

The $\epsilon_{abc_2...c_d} \omega^{ab} \wedge e^{c_2} \wedge \ldots \wedge e^{c_d}$ term in $S_{\text{bndy}}$ has a relatively simple form, but to obtain this in the way we have here is non-trivial. Instead, we could have motivated it by starting from the first-order vielbein formulation of gravity, see for example [73], in which we write the bulk portion of the action as

$$S_{\text{bulk}} = -\frac{1}{16\pi G} \int_M \frac{\epsilon_{abc_2...c_d}}{(d-1)!} \left( R^{ab} - \frac{2\Lambda}{d(d+1)} e^a \wedge e^b \right) \wedge e^{c_2} \wedge \ldots \wedge e^{c_d}, \quad (225)$$

where $R^{ab} \equiv d\omega^{ab} + \omega^a{}_c \wedge \omega^{cb}$. This is identically equal to the usual Einstein-Hilbert action. This form makes it clear that the only derivative appears in the curvature, so upon variation the boundary term is given by

$$\theta = -\frac{1}{16\pi G} \frac{\epsilon_{abc_2...c_d}}{(d-1)!} \delta\omega^{ab} \wedge e^{c_2} \wedge \ldots e^{c_d}, \quad (226)$$

which is evidently compatible with Dirichlet boundary conditions on the spin connection, not the metric/vielbein. To make the variational principle compatible with Dirichlet boundary condition on the vielbein, it would be sufficient to add a term like $\sim \epsilon_{abc_2...c_d} \omega^{ab} \wedge e^{c_2} \wedge \ldots \wedge e^{c_d}$, which is precisely the coordinate-independent term we found in our calculation of the GHY boundary term. The remaining term $\sim \partial_a n^a$ is also compatible with Dirichlet boundary conditions on the vielbein because its variation can be shown to be independent of the normal derivatives of $\delta n_a$, which could in principle have state dependence through the flat index.

Specializing now to $D = 3$, the boundary action (223) becomes

$$S_{\text{bndy}} = -\frac{1}{16\pi G} \int_{\partial M} \left[ -\epsilon_{abc} \omega^{ab} \wedge e^c + 2\sqrt{h} d^2 x \partial_a n^a \right], \quad (227)$$

so upon writing $\omega_a = \frac{1}{2} \epsilon_{abc} \omega^{bc}$ and converting to the Chern-Simons connections

$$A^a = \omega^a + e^a, \quad \overline{A}^a = \omega^a - e^a, \quad (228)$$

we find

$$-\epsilon_{abc} \omega^{ab} \wedge e^c = 2 \operatorname{tr}(A \wedge \overline{A}). \quad (229)$$

Hence, the GHY term in the Chern-Simons variables may be written

$$S_{\text{bndy}} = -\frac{1}{16\pi G} \int_{\partial M} \left[ 2 \operatorname{tr}(A \wedge \overline{A}) + 2\sqrt{h} d^2 x \partial_a n^a \right]. \quad (230)$$

It should also be noted that when transforming the bulk action into Chern-Simons variables, another factor of $\operatorname{tr}(A \wedge \overline{A})$ appears from a total derivative in the bulk action. The boundary action presented here is only equal to the GHY part, and does not include this additional contribution.

# E   Integrals

In this appendix, we review how to perform the integrals which appear in our loop computations. Starting in section E.1, we review how to perform a slight generalization of the entire class of integrals which appear in our 1-loop calculations. In section E.2, we demonstrate how to perform a class of Fourier transform within dimensionally-regularized integrals which we found useful while preparing this paper. Section E.3 displays the details of the 2-loop self-energy calculation, since this integral cannot be reduced to the integrals that appear in the 1-loop calculations. Finally, in section E.4 we perform an example calculation showing how a perturbative calculation using the propagator (142) relates to the calculation using the covariant rule (143).

## E.1   1-Loop Integrals

Here we review how to perform some of the integrals which appear in 1-loop calculations, which take the generic form

$$I_{n,m}(r;\Delta) \equiv \int \frac{d^d k}{(2\pi)^d} \frac{(k_z)^n (k_{\bar{z}})^m}{[k^2 + \Delta]^r} \,. \tag{231}$$

In the process we will also review how to perform some other standard integrals in dimensional regularization.

We understand the numerator of the integrand in (231) as a particular tensor product of momenta components, much like

$$\int \frac{d^d k}{(2\pi)^d} \frac{k_\mu k_\nu}{[k^2 + \Delta]^r} \propto \delta_{\mu\nu}, \tag{232}$$

or its generalization to an arbitrary product of components in the numerator. With this in mind, we will think of the $d$-dimensional domain of integration as containing a 2-dimensional subspace on which we choose the complex coordinates $k_z$ and $k_{\bar{z}}$. As a result, the $d$-dimensional inner product will be given by $p \cdot q = 2(p_z q_{\bar{z}} + p_{\bar{z}} q_z) + p_\perp \cdot q_\perp$ where $p_\perp$ and $q_\perp$ are the components of $p$ and $q$ orthogonal to the 2-dimensional subspace we have singled out.

This setup allows us to produce a generating function for the integrals $I_{n,m}$ by first noting the identity

$$\frac{(k_z)^n (k_{\bar{z}})^m}{[k^2 + 2p \cdot k + \Delta]^r} = \frac{\Gamma(1-r)}{\Gamma(n+m-r+1)} \left(\frac{1}{4} \frac{\partial}{\partial p_{\bar{z}}}\right)^n \left(\frac{1}{4} \frac{\partial}{\partial p_z}\right)^m \frac{1}{[k^2 + 2p \cdot k + \Delta]^{r-n-m}}, \tag{233}$$

and writing

$$\begin{aligned}
I_{n,m}(r;\Delta) &= \frac{\Gamma(1-r)}{\Gamma(n+m-r+1)} \left(\frac{1}{4} \frac{\partial}{\partial p_{\bar{z}}}\right)^n \left(\frac{1}{4} \frac{\partial}{\partial p_z}\right)^m \int \frac{d^d k}{(2\pi)^d} \frac{1}{[k^2 + 2p \cdot k + \Delta]^{r-n-m}} \Bigg|_{p_z, p_{\bar{z}}=0} \\
&= \frac{\Gamma(1-r)}{\Gamma(n+m-r+1)} \left(\frac{1}{4} \frac{\partial}{\partial p_{\bar{z}}}\right)^n \left(\frac{1}{4} \frac{\partial}{\partial p_z}\right)^m I_{0,0}(r-n-m;\Delta-p^2) \Bigg|_{p_z, p_{\bar{z}}=0},
\end{aligned} \tag{234}$$

so we can generate all the $I_{n,m}$ in terms of $I_{0,0}$ and its derivatives.

To perform the integral $I_{0,0}$ we write

$$\begin{aligned}
I_{0,0}(r;\Delta) &= \int \frac{d^d k}{(2\pi)^d} \frac{1}{(k^2 + \Delta)^r} = \int \frac{d^d k}{(2\pi)^d} \frac{1}{\Gamma(r)} \int_0^\infty dx\, x^{r-1} e^{-x(k^2+\Delta)} \\
&= \frac{1}{(4\pi)^{d/2}} \frac{\Delta^{d/2-r}}{\Gamma(r)} \Gamma(r-d/2) \,,
\end{aligned} \tag{235}$$

where in the second line we have used the identity

$$\frac{1}{\alpha^z} = \frac{1}{\Gamma(z)} \int_0^\infty dx\, x^{z-1} e^{-\alpha x} , \qquad (236)$$

and then finally performed the remaining Gaussian integral in $k$, identifying the remaining integral over $x$ as being a Gamma function.

Putting everything together, we find

$$I_{n,m}(r;\Delta) = \frac{\Gamma(r-2n-d/2)}{(4\pi)^{d/2}\Gamma(r)} \left(\frac{1}{4}\frac{\partial}{\partial p_{\bar{z}}}\right)^n \left(\frac{1}{4}\frac{\partial}{\partial p_z}\right)^m (\Delta - 4p_z p_{\bar{z}})^{d/2-r+2n}\Bigg|_{p_z, p_{\bar{z}}=0} . \qquad (237)$$

From this generating function we can also note that only rotationally invariant integrands will be non-zero. That is, $I_{n,m} \propto \delta_{n,m}$.

Since many of our diagrams have two propagators carrying momenta, the special case $I_{n,n}(2;\Delta)$ will be particularly important. These integrals can always be put into the form

$$I_{n,n}(2;\Delta) = \frac{Z_{2,n}}{(4\pi)^{d/2}} \Gamma(2-d/2)\Delta^{d/2+n-2} , \qquad (238)$$

where the coefficients $Z_{2,n}$ depend only on $d$ and $n$. The first few of these coefficients are given by

$$Z_{2,0} = 1, \quad Z_{2,1} = \frac{1}{2(2-d)}, \quad Z_{2,2} = -\frac{1}{2d(2-d)}, \quad Z_{2,3} = \frac{3}{4d(2-d)(2+d)} . \qquad (239)$$

The above integral allows us to perform all 1-loop integrations.

## E.2 Some Fourier Transforms

We have also found it useful to compute the Fourier transform in $d$ dimensions of functions with the form $k_z^m k_{\bar{z}}^n (k^2)^s$, which we find to be

$$\begin{aligned}
R^s_{m,n}(x) &\equiv \int \frac{d^d k}{(2\pi)^d} e^{ik\cdot x} (k^2)^s k_z^m k_{\bar{z}}^n \\
&= (-i)^{m+n} \frac{4^s}{\pi^{d/2}} \frac{\Gamma(s+d/2)}{\Gamma(-s)} \partial_z^m \partial_{\bar{z}}^n (z\bar{z} + x_\perp^2)^{-s-d/2}.
\end{aligned} \qquad (240)$$

To show this we will take the same approach as we did for the 1-loop integrals and obtain a generating function for them. To this end, we first perform the Fourier transform

$$\begin{aligned}
R^s_{0,0}(x) &= \int \frac{d^d k}{(2\pi)^d} e^{ik\cdot x} (k^2)^s = \int \frac{d^d k}{(2\pi)^d} e^{ik\cdot x} \frac{1}{\Gamma(-s)} \int_0^\infty d\alpha\, \alpha^{-s-1} e^{-\alpha k^2} \\
&= \frac{1}{(4\pi)^{d/2}\Gamma(-s)} \int_0^\infty d\alpha\, \alpha^{-s-1-d/2} e^{-\frac{x\cdot x}{4\alpha}} = \frac{4^s}{\pi^{d/2}} \frac{\Gamma(s+d/2)}{\Gamma(-s)} \frac{1}{(x\cdot x)^{s+d/2}} ,
\end{aligned} \qquad (241)$$

where we have performed the Gaussian integral over $k$ and reidentified the result as a Gamma function after rescaling the integration variable to $\beta = \frac{x\cdot x}{4\alpha}$.

With this, we complete our calculation by writing

$$\begin{aligned}
R^s_{m,n}(x) &= \left(\frac{1}{i}\partial_z\right)^m \left(\frac{1}{i}\partial_{\bar{z}}\right)^n \int \frac{d^d k}{(2\pi)^d} e^{ik\cdot x} (k^2)^s = \left(\frac{1}{i}\partial_z\right)^m \left(\frac{1}{i}\partial_{\bar{z}}\right)^n R^s_{0,0}(z\bar{z} + x_\perp^2) \\
&= (-i)^{m+n} \frac{4^s}{\pi^{d/2}} \frac{\Gamma(s+d/2)}{\Gamma(-s)} \partial_z^m \partial_{\bar{z}}^n (z\bar{z} + x_\perp^2)^{-s-d/2} ,
\end{aligned} \qquad (242)$$

where we have assumed $x$ to have raised index and introduced a shorthand in which the two complex coordinates of $x$ are denoted by $z$ and $\bar{z}$ so $x \cdot x = z\bar{z} + x_\perp^2$. This establishes the claimed form result of the Fourier transform.

By expanding $R_{m,n}^s$ on both sides in a power series in $s$ and matching terms, the expansion

$$(k^2)^s = \sum_{\ell=0}^{\infty} \frac{1}{\ell!} (\ln k^2)^\ell s^\ell \,, \tag{243}$$

allows us to also generate the Fourier transform of functions with the form $k_z^m k_{\bar{z}}^n (\ln k^2)^\ell$ as well. Of particular note in this paper are the special cases

$$\int \frac{d^2 k}{(2\pi)^2} e^{ik \cdot x} k_z^n k_{\bar{z}}^m \ln k^2 = -\frac{(-1)^{\frac{n+m}{2}}}{\pi} \frac{\Gamma(m+1)\Gamma(n+1)}{z^{n+1}\bar{z}^{m+1}} \,,$$
$$\int \frac{d^2 k}{(2\pi)^2} e^{ik \cdot x} k_z^n k_{\bar{z}}^m (\ln k^2)^2 = \frac{2(-1)^{\frac{n+m}{2}}}{\pi} \frac{\Gamma(n+1)\Gamma(m+1)}{z^{n+1}\bar{z}^{m+1}} \left(2\gamma - H_m - H_n + \ln(z\bar{z})^2\right) \,, \tag{244}$$

where $\gamma$ is the Euler-Mascheroni constant and $H_n$ is the $n^{\text{th}}$ harmonic number. In particular, these two integrals appear when writing the stress tensor correlator (176) in position space.

### E.3 $\langle f'(p)f'(-p)\rangle$ propagator at two-loop order

In this appendix, we compute the following integral which appears in the calculation of the propagator at two-loop order

$$I = \int \frac{d^2 k}{(2\pi)^2} \int \frac{d^2 k'}{(2\pi)^2} \frac{k_{\bar{z}}^2 k_{\bar{z}}'^2 (p_z - k_z - k_z')^2}{k^2 k'^2 (p-k-k')^2} \,. \tag{245}$$

We start by using Feynman parameters to write

$$I = \int \frac{d^2 k}{(2\pi)^2} \int \frac{d^2 k'}{(2\pi)^2} k_{\bar{z}}^2 k_{\bar{z}}'^2 (p_z - k_z - k_z')^2$$
$$\times \int_0^1 du \int_0^{1-u} dv \frac{\Gamma(3)}{\Gamma(1)^3} \frac{1}{(uk^2 + vk'^2 + (1-u-v)(p-k-k')^2)^3} \,. \tag{246}$$

We then change momentum variables, noting that

$$uk^2 + vk'^2 + (1-u-v)(p-k-k')^2 = \alpha q^2 + \alpha' q'^2 + \gamma p^2 \,, \tag{247}$$

with

$$\alpha = 1 - v \,, \quad \alpha' = \frac{(u+v)(1-v) - u^2}{1-v} \,, \quad \gamma = \frac{uv(1-u-v)}{(u+v)(1-v) - u^2} \,, \tag{248}$$

and

$$q = k + \frac{1-u-v}{1-v}(k'-p) \,, \quad q' = k' - \frac{u(1-u-v)}{(u+v)(1-v) - u^2} p \,. \tag{249}$$

The change of momenta variables from $k, k'$ to $q, q'$ has a trivial Jacobian. To continue, we convert the denominator to an exponential using a Schwinger parameter as in formula (236). Explicitly,

$$\frac{1}{(\alpha q^2 + \alpha' q'^2 + \gamma p^2)^3} = \frac{1}{\Gamma(3)} \int_0^\infty dU \, U^2 e^{-U(\alpha q^2 + \alpha' q'^2 + \gamma p^2)} \,. \tag{250}$$

We now have

$$I = \int \frac{d^2q}{(2\pi)^2} \int \frac{d^2q'}{(2\pi)^2} k_{\bar{z}}^2 k_{\bar{z}}'^2 (p-k-k')_z^2 \int_0^1 du \int_0^{1-u} dv \int_0^\infty dU\, U^2 e^{-U(\alpha q^2 + \alpha' q'^2 + \gamma p^2)}\,, \tag{251}$$

where in the first line $k$ and $k'$ are understood to be functions of $q$ and $q'$ using (249). The momentum integrals are all Gaussian of the form

$$\int \frac{d^2q}{(2\pi)^2} q_z^n q_{\bar{z}}^m e^{-U\alpha q^2}\,. \tag{252}$$

We perform these integrals via dimensional regularization. We start with the generating function

$$\tilde{G}[p,C] = \int \frac{d^d K}{(2\pi)^d} e^{-C(K^2 + 2K\cdot p)} = \frac{1}{(4\pi C)^{\frac{d}{2}}} e^{Cp^2}\,. \tag{253}$$

Noting that

$$K \cdot p = 2K_z p_{\bar{z}} + 2K_{\bar{z}} p_z + \vec{K}_\perp \cdot \vec{p}_\perp\,, \tag{254}$$

we conclude

$$\left[ \left( \frac{-1}{4C} \partial_{p_z} \right)^n \left( \frac{-1}{4C} \partial_{p_{\bar{z}}} \right)^m \tilde{G}[p,C] \right]_{p=0} = \int \frac{d^d K}{(2\pi)^d} K_z^m K_{\bar{z}}^n e^{-CK^2}\,. \tag{255}$$

This allows us to compute the integrals (252) explicitly as a function of the dimension $d$. Note in particular that this integral vanishes unless $m = n$, so really the only formula we need is

$$\left[ \frac{1}{(4C)^n} \partial_{p_z}^n \partial_{p_{\bar{z}}}^n \tilde{G}[p,C] \right]_{p=0} = \int \frac{d^d K}{(2\pi)^d} (K_z K_{\bar{z}})^n e^{-CK^2}\,. \tag{256}$$

After performing the Gaussian integrals, we find

$$\begin{aligned}
I = \int_0^1 du \int_0^{1-u} dv \int_0^\infty dU\, e^{-U\gamma p^2} &\frac{p_{\bar{z}}^2 u^2 v^2 (1-u-v)^2}{4(4\pi)^d ((u+v)(1-v)-u^2)^{4+\frac{d}{2}}} \\
\times &\left( 3U^{-d} - 8(p_z p_{\bar{z}}) U^{1-d} \frac{uv(1-u-v)}{(u+v)(1-v)-u^2} + 4(p_z p_{\bar{z}})^2 U^{2-d} \frac{v^2 u^2 (1-u-v)^2}{((u+v)(1-v)-u^2)^2} \right).
\end{aligned} \tag{257}$$

The integral over the Schwinger parameter $U$ can now be performed trivially using the formula

$$\int_0^\infty dU\, U^x e^{-U\gamma p^2} = \frac{\Gamma(1+x)}{(\gamma p^2)^{1+x}}\,. \tag{258}$$

Before performing the Feynman integrals, we expand around $d = 2 + \epsilon$. We obtain

$$I = \int_0^1 du \int_0^{1-u} dv \frac{5}{16\pi^2} \frac{u^3 v^3 (1-u-v)^3}{((u+v)(1-v)-u^2)^6} p_z p_{\bar{z}}^3 \log p^2 + \text{polynomial}\,. \tag{259}$$

Integration over Feynman parameters in Mathematica yields

$$I = \frac{1}{2^7 3\pi^2} p_z p_{\bar{z}}^3 \log p^2 + \text{polynomial}\,, \tag{260}$$

which is the formula (166) used in the main text.

### E.4 Diagram with generalized propagator

Here we provide further explanation regarding our choice of propagator, discussed below (142). Consider the following family of propagators labelled by the parameter $\eta$,

$$\langle f'(p)f'(-p)\rangle_0 = 32\pi G\left(\frac{p_z^2}{p^2} + \eta\frac{p_z p_{\bar{z}}}{p^2}\right), \quad \langle \bar{f}'(p)\bar{f}'(-p)\rangle_0 = 32\pi G\left(\frac{p_{\bar{z}}^2}{p^2} + \eta\frac{p_z p_{\bar{z}}}{p^2}\right). \quad (261)$$

Direct inversion of the quadratic terms in the action gives $\eta = 1$, while in our computations we took $\eta = 0$, claiming that this amounted to a particular Lorentz invariant renormalization scheme. To further illustrate this, we consider a typical diagram computed with general $\eta$.

In particular, consider the 1-loop contribution to $\langle f'f'\bar{f}'\bar{f}'\rangle$ in diagram (158). Using the generalized propagator, the diagram is proportional to the following integral

$$I_\eta = \int \frac{d^d k}{(2\pi)^d} \frac{k_{\bar{z}}(\eta k_z + k_{\bar{z}})}{k^2} \frac{(k_z - p_z)(k_z + \eta k_{\bar{z}} - p_z - \eta p_{\bar{z}})}{(k-p)^2}, \quad (262)$$

which, using the integral (238), we compute as

$$I_\eta = \frac{p_z p_{\bar{z}}}{8\pi\varepsilon} + \frac{p_z p_{\bar{z}}}{96\pi}\left(6\gamma - 11 + 6\ln\left(\frac{p^2}{4\pi}\right)\right) + \frac{p_z^2 + p_{\bar{z}}^2}{48\pi}\eta + \frac{p_z p_{\bar{z}}}{96\pi}\eta^2 + O(\varepsilon). \quad (263)$$

The relevant observation is that the divergent and log parts of the integral are independent of $\eta$. Furthermore, the $\eta$-dependent terms are purely polynomial in the momenta and all terms which do not respect Lorentz invariance vanish if we take $\eta$ to zero. So using a general value for $\eta$ corresponds to using a different (non-Lorentz invariant) renormalization scheme. That is, if we chose a nonzero value of $\eta$ we should also include additional non-Lorentz invariant counterterms to cancel off the non-Lorentz invariant polynomial terms in (263). A simpler way to obtain the same final result is to set $\eta = 0$ at the outset. This feature applies to all diagrams considered in this work.

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
