# Peer review of "Field Theory of Interacting Boundary Gravitons"

_SciPost Physics, doi:SciPost Phys. 13, 038 (2022)_

## Round 1 · Referee Report · Anonymous (Referee 1) · 2022-5-16

Report

This paper considers pure 3d gravity with a finite cutoff planar boundary which should be dual to a $T\bar{T}$-deformed CFT$_2$ on the line. The case where the boundary geometry is a cylinder was considered before. The planar case is simpler which allows the authors to go further than previous studies.

The authors provide an elegant derivation of the action for boundary gravitons. The result is a deformation of the Alekseev-Shatashvili action of the asymptotically AdS$_3$ case. The action is derived by examining bulk diffeomorphisms preserving a planar boundary surface at finite cutoff $r_c$. The derivation relies on an ingenious use of the momentum operator to derive the symplectic form and a field redefinition for which they assume a simple form. The boundary action is then used to compute two-point function of the stress tensor up to two loops in $G$.

The action is derived at finite value of the cutoff $r_c$ in an expansion in $G$. The renormalization procedure to compute stress tensor correlators leaves an undetermined constant $\mu$. Although the authors argue that this constant should be fixed by symmetries, they leave this for future work. An interesting conjecture by the authors is that the all order action assumes a Nambu-Goto form. Although various checks are given for this, no derivation is presented.

The paper is well written and the results are new and interesting. Although the authors leave some unresolved points, the presented work is already substantial. Therefore I recommend the paper for publication in SciPost Physics if the authors can answer the following questions:

  1. In many cases, the $T\bar{T}$ deformation can be integrated by writing down a flow equation. Why doesn't this give a way to derive the Nambu-Goto action?

  2. How to understand the Virasoro symmetries in this context? Although the Virasoro symmetries may have been addressed in previous papers, I feel like the paper would benefit from clarifying how the present work relates to previous studies. Some of the results for the circle are discussed in the introduction but the authors don't say whether they generalise to the planar case of this paper. For example the condition (3.75) seems to imply that the symplectic form is undeformed here, which appears to be in tension with the idea that the Virasoro algebra is deformed.

  3. The asymptotic AdS$_3$ case can be understood in the theory of coadjoint orbits of the Virasoro group where the constant $a$ in (3.43) labels the coadjoint orbit. The results of the authors suggest that a similar interpretation is possible at finite cutoff. It seems that understanding the group theory behind the system will clarify some of the points left unresolved by the authors and I would be curious to know if the authors have something to say about this. Is there a deformation of the theory of coadjoint orbits that underlies the results obtained here?

  4. In the asymptotic AdS$_3$ case with planar boundary, thermal states (corresponding to planar BTZ) are easy to study since they can be obtained from the vacuum by a diffeomorphism. This corresponds to a special choice of constant $a$ in (3.43) and is largely controlled by Virasoro group theory. Shouldn't the present analysis easily generalize to the planar BTZ case? In that case the SL(2) symmetry is broken to U(1) which presumably allows additional terms in (3.19).

  5. Could the authors comment on whether they expect their story to generalize to the more realistic case with matter fields in the bulk? Or is it the case, as argued in the literature, that the finite cutoff interpretation only works in pure gravity?

Requested changes

  • Note the small typo in (3.75) where the first equality should not be there.

  • validity: top
  • significance: high
  • originality: high
  • clarity: top
  • formatting: excellent
  • grammar: excellent

Author:  Ruben Monten  on 2022-05-21  [id 2500]

(in reply to Report 1 on 2022-05-16)

We would like thank the referee for their thorough review and comments. We will correct the typo in v2 and reply here to each question in turn:

  1. The flow equation can indeed be (and has been) used to identify the Nambu-Goto action as the $T \bar T$ deformation of the 2d free boson. This field theory is related, through a conjecture by McGough et al. [1611.03470], to AdS$_3$ gravity with a finite-redshift boundary. The starting point in our paper is the latter, and the fact that we find the Nambu-Goto action can be seen as evidence in support of the conjectured duality. Our derivation identifies the precise gravitational degrees of freedom that become the scalar in the Nambu-Goto action: they are the interacting boundary gravitons after a rather non-trivial and indeed non-local field redefinition.

  2. The deformation of the charge algebra can be traced back to the fact that the stress tensor gets deformed. Invariance of the symplectic form is not sufficient to guarantee preservation of the Virasoro algebra. There are state-dependent coordinate transformations [1809.09593] and canonical transformations [2001.03563] that map the deformed to the undeformed theory. It may be possible to use these to define a different slicing of phase space, and a different set of charges that preserve the Virasoro algebra. To our knowledge this question remains to be settled definitively. With regards to the difference between the plane and the cylinder: on compact space the Virasoro generators $L_n$ are labeled by discrete Fourier modes, as they must have the correct periodicity. These constraints disappears on the plane, and it is less natural to consider the $L_n$ generators.

  3. The question of a symmetry group structure underlying the $T \bar T$ deformed theory is a very interesting one. It should be pointed out that, for infinitesimal transformations, the algebra found in [2103.13398] is non-linear and hence not a Lie algebra. There seems to be no immediately apparent group that resolves this algebra, or even a guarantee that such a group exists. As such, this question falls outside the scope of our current work.

  4. The analysis should indeed extend straightforwardly to the planar BTZ geometry, at least in the metric formalism. The argument in Section 3.4 will indeed carry over, since it does not require SL(2) x SL(2) invariance.

  5. If massive matter fields are added, the present theory still describes their fluctuations around the vacuum state. This is because the matter fields can be integrated out and the resulting higher-derivative contributions in the action can be absorbed in a field redefinition of the metric by the universality argument presented by Witten [0706.3359]. This argument fails, however, if the matter fields are massless or if a nonzero classical background for the matter fields is turned on. One direct way to see this is that the trace equation is modified if the matter fields have a nonzero profile on the boundary.

Anonymous on 2022-05-24  [id 2516]

(in reply to Ruben Monten on 2022-05-21 [id 2500])

I would like to thanks the authors for their clarifying replies and recommend proceeding with the publication.

---

## Round 1 · Referee Report · Daniel Harlow (Referee 2) · 2022-7-4

Report

This is a nice paper doing a two-loop calculation of the boundary stress tensor two-point function in $AdS_3$ gravity with a finite cutoff. I think it should be published essentially as is, but I have a few related pedagogical comments on the covariant phase space section of the paper:

1) On page 10 the authors give an incomplete characterization of what it means for a diffeomorphism generator to preserve the boundary conditions. Namely they are working in a gauge where the boundary location is always at $r=r_c$, and to preserve this they need to require that $\xi^\mu$ is tangent to this surface. Otherwise the vector field $V_\xi$ does not preserve pre-phase space since it evolves solutions which obey the boundary conditions to solutions which do not.

2) The authors comment that equation 3.14 holds even when $\xi^\mu$ is not tangent to the boundary. This is true, but it is an artifact of working Einstein gravity. Namely in Einstein gravity one can choose a gauge where the quantity $C$ defined in reference 24 vanishes, and the Gaussian normal coordinates 3.3 are such a gauge. More generally the derivation of 3.14 requires an assumption that $C$ is covariant under diffeomorphisms generated by $\xi^\mu$, and when $C$ is nonzero (as it would be e.g. once higher-derivative terms are included) then this likely requires $\xi^\mu$ to be tangent to the boundary.

3) Even in Einstein gravity, although in certain gauges 3.14 does not require $\xi^\mu$ to be tangent to the boundary, the conservation of the symmetry charge certainly requires this. So any $\xi^\mu$ which is not tangent to the boundary does not define a flow which preserves the phase space, is not generated by a conserved charge, and should not be viewed as a symmetry. It may still be useful to consider (in fact in section 3.7 of reference 24 we did an analysis of JT gravity which is similar to what is done here), but for my money it would be good to be a bit more clear about what is going on and in particular what are symmetries and what are not.

4) I found the notation $Q[\xi]$ a bit confusing, as it resembles Wald's (now standard) notation $Q_\xi$ for what he calls the Noether charge. In fact the $Q[\xi]$ here is indeed the thing that SHOULD be called the Noether charge (what was Wald thinking?), but as this notation is standard it might be worth making a brief clarification that $Q[\xi]$ here is not Wald's $Q_\xi$.

None of these comments threaten the results of the paper, which I am happy to recommend for publication, but I hope they are useful for the authors.

PS Sorry for the delay on this report!

  • validity: high
  • significance: good
  • originality: good
  • clarity: high
  • formatting: excellent
  • grammar: excellent

Author:  Ruben Monten  on 2022-07-16  [id 2664]

(in reply to Report 2 by Daniel Harlow on 2022-07-04)

Thanks Daniel for the constructive feedback!

We agree on points 2,3,4 and will add comments to that effect in v2, around equation 3.12 and in footnote 13. Regarding point 2, we note that within our general context of pure 3d gravity the restriction to Einstein gravity is not really a restrictions at all, since any (parity invariant) action with higher derivatives can, at least order by order, be put back in standard 2-derivative form by a field redefinition. Going beyond pure 3d gravity would change many aspects of the analysis, and the need for a nonzero $C$ is one of these.

On point 1, we would like to stress that it is essential that we do not restrict to diff vector fields that are tangent to the boundary. A more geometrical characterization of our phase space is that of all flat surfaces embedded in an ambient AdS$_3$. To translate this picture into the one used in the draft, we note that near each such surface we can construct a Gaussian normal coordinate system, with the surface at $r = r_c$. The coordinate transformation needed to relate two such surface then clearly requires diffs that are not tangent to the surfaces. The only place where such a tangency restriction would be imposed is in identifying charges which are conserved under time evolution. To clarify these issues we are adding this explanation between equations 3.11 and 3.12 in the new version of the paper.

---

## Round 2 · List of Changes

• We added comments and clarifications to address the referees' remarks,
  • added references,
  • and fixed typos.

---

## Editorial Decision

published